# Cell Surface B2m-Free Human Leukocyte Antigen (HLA) Monomers and Dimers: Are They Neo-HLA Class and Proto-HLA?

**DOI:** 10.3390/biom13081178

**Published:** 2023-07-28

**Authors:** Mepur H. Ravindranath, Narendranath M. Ravindranath, Senthamil R. Selvan, Fatiha El Hilali, Carly J. Amato-Menker, Edward J. Filippone

**Affiliations:** 1Department of Hematology and Oncology, Children’s Hospital, Los Angeles, CA 90027, USA; 2Terasaki Foundation Laboratory, Santa Monica, CA 90064, USA; 3Norris Dental Science Center, Herman Ostrow School of Dentistry, University of Southern California, Los Angeles, CA 90089, USA; nravindr@usc.edu; 4Division of Immunology and Hematology Devices, OHT 7: Office of In Vitro Diagnostics, Office of Product Evaluation and Quality, Center for Devices and Radiological Health, Food and Drug Administration (FDA), Silver Spring, MD 20993, USA; senthamil.selvan@fda.hhs.gov; 5Medico-Surgical, Biomedicine and Infectiology Research Laboratory, The Faculty of Medicine and Pharmacy of Laayoune & Agadir, Ibnou Zohr University, Agadir 80000, Morocco; f.elhilali@uiz.ac.ma; 6Department of Microbiology, Immunology and Cell Biology, School of Medicine, West Virginia University, Morgantown, WV 26506, USA; carly.j.amato@gmail.com; 7Division of Nephrology, Department of Medicine, Sidney Kimmel Medical College, Thomas Jefferson University, Philadelphia, PA 19145, USA; kidneys@comcast.net

**Keywords:** human leukocyte antigens, beta2-microglobulin, heavy chains, monomers, homodimers, heterodimers

## Abstract

Cell surface HLA-I molecules (Face-1) consist of a polypeptide heavy chain (HC) with two groove domains (G domain) and one constant domain (C-domain) as well as a light chain, B2-microglobulin (B2m). However, HCs can also independently emerge unfolded on the cell surface without peptides as B2m-free HC monomers (Face-2), B2m-free HC homodimers (Face 3), and B2m-free HC heterodimers (Face-4). The transport of these HLA variants from ER to the cell surface was confirmed by antiviral antibiotics that arrest the release of newly synthesized proteins from the ER. Face-2 occurs at low levels on the normal cell surface of the lung, bronchi, epidermis, esophagus, breast, stomach, ilium, colorectum, gall bladder, urinary bladder, seminal vesicles ovarian epithelia, endometrium, thymus, spleen, and lymphocytes. They are upregulated on immune cells upon activation by proinflammatory cytokines, anti-CD3 antibodies, antibiotics (e.g., ionomycin), phytohemagglutinin, retinoic acid, and phorbol myristate acetate. Their density on the cell surface remains high as long as the cells remain in an activated state. After activation-induced upregulation, the Face-2 molecules undergo homo- and hetero-dimerization (Face-3 and Face-4). Alterations in the redox environment promote dimerization. Heterodimerization can occur among and between the alleles of different haplotypes. The glycosylation of these variants differ from that of Face-1, and they may occur with bound exogenous peptides. Spontaneous arthritis occurs in HLA-B27+ mice lacking B2m (HLA-B27+ B2m−/−) but not in HLA-B27+ B2m+/− mice. The mice with HLA-B27 in Face-2 spontaneous configuration develop symptoms such as changes in nails and joints, hair loss, and swelling in paws, leading to ankyloses. Anti-HC-specific mAbs delay disease development. Some HLA-I polyreactive mAbs (MEM series) used for immunostaining confirm the existence of B2m-free variants in several cancer cells. The upregulation of Face-2 in human cancers occurs concomitantly with the downregulation of intact HLAs (Face-1). The HLA monomeric and dimeric variants interact with inhibitory and activating ligands (e.g., KIR), growth factors, cytokines, and neurotransmitters. Similarities in the amino acid sequences of the HLA-I variants and HLA-II β-chain suggest that Face-2 could be the progenitor of both HLA classes. These findings may support the recognition of these variants as a neo-HLA class and proto-HLA.

## 1. Introduction

Cell surface Human Leukocyte Antigen (HLA) class-I molecules are heteromeric, peptide-binding glycoproteins comprised of (i) a glycosylated heavy chain (HC) polypeptide (44 kDa), and (ii) a non-glycosylated free single IgSF C-like domain protein, beta 2-microglobulin (B2m, 11.6 kDa) [1,2]. The HC consists of two membrane-distal domains, designated by IMGT labels [3,4] as G (groove) domains (G-ALPHA1 and G-ALPHA2 or D1, D2) of the major histocompatibility superfamily (MhSF), and a membrane-proximal constant domain (C-like or D3) of the immunoglobulin superfamily (IgSF). The heteromer is assembled in the endoplasmic reticulum (ER) and transported to the cell surface. The membrane-distal α1 and α2 domains form a groove that binds to a peptide of 8–11 amino acids (a.a). B2m stabilizes the conformational orientation of the HC groove after the emergence from the ER by associating with the α3 domain and restricts the peptide length in the groove [1]. Notably, the peptide binding groove in HLA class-II molecules that lack B2m can accommodate longer peptides (12–15 a.a). The HLA-I trimers (HC + B2m + bound peptide) present the peptide antigen to CD8+ Cytotoxic T Lymphocytes CTLs [5]. The isotypes of HLA-I, namely classical HLA-A, HLA-B, and HLA-C, and non-classical HLA-E, HLA-F, and HLA-G, are diagrammatically illustrated in Figure 1.

HLA class II molecules differ strikingly from HLA-I by having two heavy chains (α chain and β chain, designated as II-ALPHA chain and II-BETA chain by IMGT labels). Each alpha and beta chain comprises groove (G-ALPHA [D1] and G-BETA [D1]) and constant (C-like ALPHA [D2] and C-like BETA [D2]) domains [3,4]. They are non-covalently associated to generate a conformationally stable binding site for a peptide. The structure of HLA-II illustrates that a dimerization of two HCs can provide structural stability and a stable site for binding. The genes encoding the heavy chains of these two classes of HLA are closely linked to each other on the short arm of chromosome 6, whereas the gene that encodes B2m is on chromosome 15.

Cell surface HLA-I molecules are capable of binding to other proteins to exert a signal-transducing function [6,7]. Due et al. [6] minimized the binding of insulin receptor to the cell surface HLA-I using HC-specific monoclonal antibodies (mAbs) (PA 2.6 and BB 7.5, -6, and -7). However, the B2m-specific polyclonal Abs did not affect such binding of insulin receptor to HLA, which suggests that the cell surface HLA HCs can independently serve as ligands without the involvement of B2m. This finding initiated the study of “empty HLA” molecules (peptide-free and B2m-free) now referred to as open conformers by Arosa et al. [8]. Since the term “open conformers” may imply both the dimeric, peptide-free B2m-associated HC and the monomeric peptide-free B2m-free HCs, the term Face-2 was coined to distinguish B2m-free HC variants from peptide-free B2m-associated HCs and B2m, HC, peptide HLA trimers (Face-1) [9].

Early (pre-1980) HLA researchers believed that HC could not exist on the cell surface devoid of B2m, based on observations made on mutant cell lines. However, knowledge of the structural variants and their functional potentials has been steadily emerging since then. It has become increasingly clear that B2m-free HCs (Face-2) have a highly restricted half-life as they dimerize to form homo- and hetero-dimeric variants (Face-3 and Face-4) and perform diversified ligand-receptor functions [10].

The objective of this review is to chronologically trace (1) the discovery of B2m-free monomeric variants of HLA (Face-2) in normal cells, transfected cells, malignant cells, and activated immune cells; (2) the research documenting the cell surface expression of Face-2 independently of the B2m-associated HLA trimer (Face-1), as well as homo- and hetero-dimerization of Face-2 (Face-3 and Face-4); (3) the progressive research on the differences in the glycosylation patterns between Face-1 and Face-2; their peptide-binding capabilities; (4) their interactions with ligands such as KIR, hormones, and neurotransmitters; (5) their involvement in up- and downregulation of immune functions; (6) the possible role these variants in the pathogenesis of spondyloarthritis, human cancers, and in allograft recipients; and (7) the research on the phylogenetic relationship between HLA class I and II and the similarities of the amino acid patterns of the variants with other classes of HLA, which sheds new light on the origin of HLA during vertebrate evolution. The findings in this review not only highlight the need to study immune responses to the variants, but also suggest that the different Faces of HLA variants could constitute a new class of HLA. The ultimate question is: should the novel B2m-free HLA HC variants (Faces-2, -3, and 4) that emerge directly from the ER, independently of Face-1, be considered as a neo-HLA class(es)?

## 2. Early Reports on the Formation of Cell Surface B2m-Free HCs (Face-2)

### 2.1. Are Cell Surface B2m-Free HCs Ephemeral Due to the Dissociation of B2m?

In Face-1 HLA trimers, the association of B2m with HC increases the affinity of HC for peptides. Similarly, B2m increases the stability of the peptide on the groove of the HC [11,12,13,14,15,16,17]. In a unique experiment, Demaria et al. [11] selectively removed any B2m-free HCs (Face-2) on the cell surface of PMA-activated T cells with brief treatment of cold trypsin. This treatment did not remove B2m-associated HCs on the cell surface. They exposed the PMA-treated resting T cells and trypsin-treated PMA-activated cells to BFA, which arrests the exit of newly synthesized proteins from the ER. Interestingly, the BFA blocked the expression of Face-2 on resting T cells exposed to PMA but did not block the reappearance of Face-2 on trypsin-treated cells, documenting the dissociation of B2m from intact Face-1 molecules. The dissociation of B2m results in the unfolding of α1 and α 2 domains of HCs; these are referred to as “non-conformed” B2m-free HCs. These “non-conformed” B2m-free HCs are rather distinct from folded α1 and α2 domain bearing HCs or “conformed” HCs, as is observed on cells normally expressing functional B2m [17]. Notably, such peptide-carrying conformed B2m-free HCs are also observed in B2m-deficient mice [17]. Figure 2 illustrates the dissociation of B2m from the B2m-associated intact HLA (Face-1), as well as the consequent release of peptides and the formation of non-conformed membrane-bound HCs, which are selectively cleaved by membrane metalloproteinases (MMPs) [12,13,14]. Hence, the cell surface non-conformed B2m-free HCs is considered to have a very insignificant half-life. Figure 2 is based on the findings and the illustrations of Elliott [14] and Demaria [12].

The proposition that HLA-I HCs cannot be expressed on the cell surface without B2m is based on the unique properties of two cell lines. The first is Daudi, the Burkitt Lymphoma cell line. It lacks the ability to synthesize B2m and cannot express trimeric HLA-I molecules (Face-1) on the cell surface [20]. The surface expression of Face-1 in the cell line was achieved by supplementing either with mouse or human B2m. The second is the R1 cell line of a somatic cell variant of the C3H (H-2^k^) thymoma [21].

### 2.2. Can HLA HCs Emerge on the Cell Surface as B2m-Free HCs?

The first indication for the presence of B2m-free HCs of HLA-I on the cell surface emerged from the works of Krangel et al. [22] in 1979. They observed two antigenically distinct populations of HLA-A (A1 and A2) and HLA-B (B8 and B27) HCs on the surface of a human lymphoblastoid cell line T5-1. One HC population is associated with B2m (mAb W632 reactive) and another HC population without B2m, mAb W6/32 non-reactive but positive from anti-HC serum (anti-H) that was raised against HLA-B7 B2m-free HCs. *Notably, these two categories differed in their glycosylation (mannose residues), density, and location on the cell surface*. Indeed, the anti-H-positive HCs are detectable on the surface of the T5-1 cell line, although their origin is not clarified. They may be a result of the movement of anti-H-positive HCs from locations inside the cytoplasm to the cell surface. Their presence is more easily accounted for, however, by the dissociation of B2m from the B2m-associated HCs once they reach the cell surface. This notion is plausible, since the dissociation of such complexes can be detected in cell lysates at 37 °C.

In the mouse B6 lymphoma EL4 cell line, Potter et al. [23,24] observed a variant which, in contrast to the wild-type cell line, failed to express B2m on the cell surface and expressed only Face-2 (H-2D^b^). This was identified by serological reactivity as well as by CTLs. *They speculated that alterations in the glycosylation site on the α3 domain may have prevented the association of B2m to the HC (H-2D^b^).* Allen et al. [25] noted a high level of expression of the H-2D^b^ HC at the cell surface of the EL4 cell line even though there is no B2m protein within the cell. This was further confirmed by transfecting the H-2Db gene into the RIE tumor cell line, which lacks B2m. Interestingly, the D^b^ B2m-free HC on the surface of RIE/D^b^ cells failed to react with mAb specific for the α1 domain (tested with mAb B22-249.1) or the α2 domain, *suggesting an alternative structural configuration of Face-2* [15,25]. However, in stark contrast to these reports, Bix and Raulet [17] observed that the D^b^ B2m-free HCs on the surface of RIE/D^b^ cells were reactive to mAbs B22-249-1 and 28-14-8 D^b^ and suggested that *peptides can bind to HCs in the absence of B2m “with sufficient affinity to establish a functionally conformed molecule on the cell surface (p.833)”.*

**Inference:** These studies unravel differences in the glycosylation patterns of Face-1 and Face-2. Such alterations in glycosylation sites may have prevented HC association with B2m and led to the hypothesis that there is independent surface expression of Face-2, without dissociating from B2m.

## 3. Research on B2m-Free HLA Variants on Tumor Cells

### 3.1. Do Proinflammatory Cytokines Induce the Expression of Face-2 on Tumor Cells?

On a human melanoma cell line Colo-38, Giacomini et al. [26] used three different mAbs (W6/32, NAmb-1, an IgG1 directed against B2m, and Q1/28, an IgG2b that recognizes a monomorphic determinant on B2m-free HC) and examined the impact of human recombinant IFN-γ. The IFN-γ-treated cells were suspended in a medium containing ^35^S-methionine and immunoprecipitated with the above mAbs. Examining the immunoprecipitates of IFN-γ-treated and untreated cells by SDS-PAGE, they found a greater increase in the synthesis of HLA B2m-free HCs (Face-2) in IFN-γ-treated cells than in untreated cells; a similar greater increase in treated cells was not found for intact HLA (Face-1). Since the cell surface of human [27] and mouse [23,24] cells were stained with anti-sera raised against B2m-free HCs, it was concluded that *the proinflammatory cytokine IFN-γ induced the surface expression of B2m-free HCs (Face-2) without any involvement of B2m-associated HCs (Face-1).*

### 3.2. Post-Translational Expression of Face-2 Independent of Face-1 in Cancer Cells

In 1986, Bushkin et al. [28,29] observed a close association between Face-2 and a T-cell receptor molecule in patients with T cell chronic lymphocytic leukemia. While characterizing the α/β T cell receptor molecules on the leukemia cells, a mAb A1.4 (IgG 1) recognizing B2m-free HCs of HLA-A, -B, and -C molecules was used to immunoprecipitate the HCs from the lysates of the leukemia cells. Not only was a 43 kDa HC precipitated without B2m by A1.4, but a novel 38 kDa molecule was as well. Similarity between the β chains of the T cell receptor and the 38 kDa molecule was confirmed by endo-F digestion, 2D (IEF-SDS-PAGE) electrophoresis, and chymotryptic peptide mapping. Furthermore, it was confirmed that the 38 kDa molecule expressed on the surface of the leukemia cells is non-covalently associated with B2m-free HLA–HCs, detected by mAb AI.4. Evidently, B2m-free HLAs can associate with proteins other than B2m, as reported earlier by Due et al. [6]. However, these reports failed to clarify whether B2m-free HLA-I HC expression is the result of dissociation of B2m, as shown in Figure 1, or whether B2m-free HCs reach the cell surface independently.

To address this dilemma, Marozzi et al. [30] evaluated the expression of Face-2 molecules in two different human neuroblastoma (NB) cell lines, IMR-32 and LA-N-1. These cell lines express very low levels of Face-1 molecules, while detectable levels of Face-2 were noted in about 5% of cells. The treatment of IMR-32 cells with retinoic acid (RA) induced the cells to differentiate with morphological changes. RA-treated cells were tested with mAb W6/32 and with the mAb L31 specific for B2m-free HCs. Grassi et al. [31] characterized this murine mAb, finding that it binds to Face-2, and noted that it binds to an epitope on the α1 domain of the HC, particularly to the tyrosine or phenylalanine at position 67. All HLA-C alleles (C1 to C8) and a small group of B alleles share this specific aromatic a.a. at position 67 (Y^67^ & F^67^). Upon further cellular differentiation, the percentage of cells expressing L31-reactive molecules on the cell surface increased from 45 to 80%.

The same analysis was carried out on the NB cell line LA-N-1. On RA-untreated cells, both W6.32 and L31 gave results comparable to those shown by IMR-32 cells. An RA-treated LA-N-1 cell population showed a 30% increase in mAb L31 reactivity, while the reactivity of mAb W6/32 did not change from the untreated level. The exposure of B2m-free HCs on the surface of the differentiated NB cells was not accompanied by significant changes in the level of intact HLA. The synthesis of intact HLAs or B2m mRNAs or of L31 proteins did not change in differentiated NB cells, suggesting that the surface expression of B2m-free HCs is regulated post-translationally.

In contrast to the contention that B2m-free HCs emerge from the dissociation of B2m-associated HLA, the investigations of Martayan et al. [32] shed new light on cell surface B2m-free HCs. They compared the conformation and surface expression of free HLA-C1 HCs in the absence of B2m in the kidney carcinoma cell line KJ29, which carries two apparently normal copies of chromosome 6 and a single chromosome 15, implying the loss of one copy of the B2m gene. They observed that the size of the pool of B2m-free HLA-CW1 molecules decreases by the availability of B2m. They concluded that “essentially all intracellular and approximately two thirds of cell-surface-expressed free HLA-CW1 HCs originate from incomplete assembly with B2m, and not from the “melting” (dissociation) of pre-formed trimeric complexes (p. 31)”.

Furthermore, the Giacomini group [33] carried out a detailed investigation by isoelectric focusing to distinguish HCs of HLA-C in complex mixtures of immunoprecipitated HLA-I molecules of two melanoma cell lines (SK Mel 37 and Colo 38) and two carcinoma cell lines (colon HT-29 and bladder T24) genotyped HLA-A, HLA-B, and HLA-C. They also performed flow cytometric analysis on viable cell suspension for studying the expression and susceptibility to IFN-γ upregulation of mAb L31 reacting HLA-C B2m-free HCs (Face-2) and compared with that of mAb W6/32. While they noted HLA-C B2m-free HCs both intracellularly and on the cell surface, IFN-γ enhanced the expression level of W6/32-reacting HLA-A, -B -C molecules (1.3- to 1 5-fold), depending on the specific allele and cell line) and L31-reacting free HLA-C heavy chains (1.5- to 12-fold) to a similar extent. *All these studies validate the hypothesis that Face-2 can emerge from cytoplasm directly and is not necessarily due to dissociation of B2m from Face-1.*

### 3.3. Does Face-2 Replace the Loss of Face-1 in Human Cancers?

Garrido [34], in a book titled *“MHC class-I loss and cancer immune escape*”, categorized the loss of HLA-Ia in human cancer into several groups, as follows:
Type 1:Tumor cells can lose all of the six HLA-I alleles present in the normal cells;Type 2:Tumor cells can lose a single HLA haplotype or one set of HLA-Ia genes localized on chromosome 6 (HLA-A, B, and C);Type 3:Tumor cells can downregulate an HLA-A, B or C locus, producing a phenotype with only four HLA-Ia alleles,Type 4:Tumor cells can lose a single HLA-I allele out of the six expressed by somatic cells;Type 5:Tumor cells can lose HLA-Ia (classical) alleles and upregulate the expression of HLA-Ib (non-classical) haplotypes, namely HLA-E, HLA-F, and HLA-G), primarily based on the correlation between the loss of HLA-Ia, concomitant with the upregulation of HLA-Ib.

The upregulation of HLA-Ib in cancer cells is primarily based on very extensive documentation of HLA-E. Interestingly, most of these reports documenting HLA-E have used the well-known and commercially available MEM series of anti-HLA-E mAbs (MEM-E02, MEM-E06, MEM-E07, MEM-E08) and mAb 3D12, which were, in fact, generated against either Face-2, or peptide-free Face-1 of HLA-E [35,36]. The positive staining by these mAbs is considered as evidence for the upregulation of HLA-E, concomitant with the loss of HLA-Ia.

The epitope specificity of both MEM-series mAbs and 3D12 [37,38] was not verified before commercializing the mAbs. When we analyzed the epitope specificity of the so-called “HLA-E specific” mAbs [39,40,41,42,43,44], using a Luminex multiplex bead system coated with different classical HLA-I Face-1 molecules admixed with Face-2 [45,46], we observed (Table 1) that these mAbs were not specific for B2m-associated (Face-1) or to B2m-free HCs (Face-2) of HLA-E.

All MEM series mAbs, most notably the extensively used mAb MEM-E/02, bound only to B2m-free HCs of several alleles of HLA-A, HLA-B, and HLA-C coated on the beads [37,38] (Table 1). It is evident that MEM series, and more particularly MEM-E/02, bind to the Face-2 of all HLA-Ia on the bead sets that are coated both with Face-1 and Face-2.

The binding of these mAbs, and particularly that of MEM-E/02, was selectively inhibited by two peptides commonly shared by almost all HLA loci. They are ^115^QFAYDGKDY^123^ (present in all alleles of HLA-A, -B, –C, -E, -F, and -G loci) and ^137^DTAAQI^142^ (present in HLA-B and HLA-C) [32] (Figure 3). Further inhibition experiments carried out with the peptide sequences on the binding of mAbs MEM-E02, E07, and E08 to HCs of different HLA loci coated on solid matrix (LABScreen beadsets) revealed the sequences (^115^QFAYDGKDY^123^ and ^137^DTAAQI^142^) and inhibited the binding of the mAbs in a dosimetric manner (Figure 3A–C in [37]).

In contrast, as shown in Table 1, the HLA-E-specific mAb (TFL-033) [40,41,42,43] did not react with any of the HLA class I molecules. On the other hand, there are several mAbs generated from the B2m-free HCs of HLA-E that were HLA-I-polyreactive [47,48], very similar to the MEM mAb series. These observations and other findings only imply that these tumor tissues and cells express B2m-free HCs (Face-2) of HLA-Ia and HLA-Ib but not specifically HLA-E. Similarly, the observations made on another mAb 3D12, claimed to be specific for HLA-E, are presented in greater detail elsewhere [38]. In light of these findings, we revisited the literature on HLA-E expression on human cancers and summarized the reports that used the MEM series of mAbs in Table 2. Evidently, the loss of HLA-I in human cancer claimed in the literature implies the loss of intact (Face-1) HLA-I molecules, but not the loss of Face-2. Garrido’s categorization of cancer types based on the loss of HLA class I molecules should be reexamined for the expression of Face-2. Evidently, the claim that HLA-Ib is upregulated concomitantly with the downregulation of HLA-Ia should be modified as HLA-Ia Face-2, and possibly Face-3 and Face-4 variants are upregulated concomitantly with the downregulation of HLA-Ia Face-1 on cancer cells.

**Inference:** The research conducted with cancer cells reveals that (1) the proinflammatory cytokine IFN-γ and RA are capable of inducing the surface expression of Face-2 without any involvement of B2m; (2) Face-2 can emerge from the cytoplasm directly and not by dissociation of B2m from Face-1; (3) the downregulation of HLA-Ia Face-1 on human cancer cells occurs concomitantly with the upregulation of Face-2 (Table 2).

## 4. Is Cell Surface Expression of Face-2 Common on Activated and Virus-Transformed Immune Cells?

Schnabl et al. [84] were the first to confirm the cell surface expression of Face-2 on activated T lymphocytes. Using mAb LA45 that reacted with B2m-dissociated denatured HLA-A, B, and C molecules but not with intact HLA-Ia, they documented that the T-cells activated in vitro or in vivo, but not those resting, expressed Face-2. Indeed, the molecular mass (45 kDa) of the antigen reacting to mAb LA45 paralleled with the expression of Face-2. Neither the anti-HLA-I mAb (W6/32) nor anti-B2m mAb recognized the LA45 positive antigen. Cloning and sequence analysis revealed a high homology of LA45 antigen cDNA and protein sequence to human HLA-I genes, characteristic of HLA-A loci. LA45 antigens could not be detected on the surface and the cytoplasm of resting peripheral blood lymphocytes (PBLs) or bone marrow mononuclear and polymorphonuclear cells. However, after the stimulation of T cells with phytohaemagglutinin (PHA), LA45 was antigen-expressed on the cell surface within 24 h, confirming the expression of HLA HCs on activated but not on resting T cells. Madrigal et al. [85] further noted “that LA45 reactivity is the property of a subpopulation of HLA-A and HLA-B molecules produced in PHA-activated T and EBV-transformed B cells but not in resting cells” (p1093). They documented that the specific epitope of HLA-I HC recognized by the mAb LA45 include arginine at position 62 and asparagine at position 63 in the α1 domain, and that the epitope is found in most of the alleles of HLA-A and B loci. As noted earlier by Schnabl et al. [84], the presence of such molecules on resting T cells is undetectable but increases from 25 K to 60 K molecules.

Demaria et al. [11] induced cell surface expression of Face-2 on resting human T cells by phorbol myristate acetate (PMA) for varying time periods and monitored its expression using Face-2-specific mAbs HCA2 and HC10 and mAb W6/32 that recognize not only Face-1, but also “B2m-free HCs (Face-2) of most HLA-B antigens” [86,87] by immunostaining and flow cytometry. The expression of Face-2 occurred prior to the expression of the interleukin-2 receptor (IL-2R). Similarly, the activation of resting T cells with anti-CD3 mAb and PHA resulted in the expression of Face-2 on average from 30 to 65% of T cells. Furthermore, the expression levels of Face-2 correlated with the levels of activation as determined by the IL-2R expression. They studied the effect of BFA (brefeldin A), which arrests the exit of newly synthesized proteins from the ER [88], on the expression of B2m-free class I heavy chains. Resting T cells were activated with PMA and stained for B2m-free class-I HCs with mAb HC, and 10:90% were positive. “Expression of these molecules was completely inhibited by BFA and thus induction of B2m-free class-I heavy chains on activated T cells requires the transport of newly synthesized MHC class I molecules to the surface (p. 107, 11)”. This report confirms that Face-2 molecules are transported from the ER to the cell surface, in contrast to the previous reports which suggested that Face-2 may be a consequence of dissociation of B2m from intact Face-1 molecules.

Interestingly, Benjamin et al. [89] also observed the presence of B2m-free HCs of HLA-B27 bound with influenza peptides on the surface of human HLA-B27-transfected B cell lines (HLA-B27-CIR). Setini et al. [90] demonstrated the presence of the Face-2 variant on EBV-transformed B-lymphoid cells using another mAb L31. They also noted that *the oligosaccharides associated with the B2m-free HCs of HLA-C are non-sialylated in contrast to B2m-associated HCs*. This is another salient finding to point out that Face-2 differs from Face-1 HCs in the glycosylation pattern.

**Inference:** The activation of resting T cells with cytokines, anti-CD3 mAb, and PHA induced the cell surface expression of Face-2. Using BFA to arrest the ER release of newly synthesized proteins, the transport of Face-2 from the ER to the cell surface was abrogated on the PMA-activated T cells. These findings clarify that Face-2 could be expressed independently of Face-1 expression on the cell surface and that is capable of binding to peptides. Furthermore, the glycosylation pattern of Face-2 may differ from that of Face-1.

## 5. Normal Tissues Express Low Levels of Face-2

Giacomini et al. [33] using mAb L31 documented a low level of cell surface expression of Face-2 of HLA-C on normal epidermis, breast, lung, bronchi, esophagus, stomach, ilium, colorectum, gall bladder, urinary bladder, seminal vesicles, ovarian epithelia, endometrium, and thymus tissues. The positivity of mAb L31 in normal tissues is comparable to that observed on carcinomas and adenocarcinomas of various grades of these tissues. As a control, they also used mAb W6/32, and, like almost all other investigators in the field of HLAs, contend that the mAb is specific for B2m-associated HCs of HLAs. However, a critical report of Tran et al. [86] questioned the contention and showed that the mAb W6/32 “actually recognizes an epitope present on isolated non-reduced α-chains of most HLA-B allelic forms”. Indeed, the report of Martayan et al. [87] on the affinity of the mAb W6/32 for HLA HCs of B2m-defective Daudi cells confirms the above contention. Although the affinity of W6/32 for Face-2 of HLA-B questions the validity of using mAb W6/32 to validate the expression of B2m-associated HCs of HLAs, particularly in the light of the observations of Giacomini et al. [26,33] that confirmed the presence of Face-2 among HLA-C alleles and some alleles of HLA-B in normal and malignant non-lymphoid human tissues and on the cell surface at low levels. In addition, a number of investigators [91,92,93,94] have documented a low level of expression of Face-2 on the cell surface of normal peripheral blood lymphocytes and spleen cells, in addition to EBV-transformed lymphoblastoid cell lines, Jeshom and Horn2 G-B27 (line 11.9) and L-B7 (lines 11.13.12) cells, and Ltk- fibroblast lines, using the mAb HC10.

**Inference:** The expression of Face-2 is not restricted to activated-immune cells, transformed cells, and cancer cells, but it is also found on the surface of normal cells at very low levels.

## 6. Monomeric and Dimeric Variants among Non-Classical HLA Molecules

The existence of Face-2 of the HLAs is further strengthened by the studies conducted on HLA class Ib loci, namely HLA-E, HLA-F, and HLA-G. HLA-G mRNA was reported in a wide variety of human tissues including peripheral blood T and B cells, keratinocytes, fetal and adult eyes, fetal thymus, liver, and male germinal cells [95]. HLA-G was reported on placenta extravillous cytotrophoblasts, which include proliferative and invading extravillous trophoblasts, endovascular interstitial trophoblasts, placental giant cells and chorionic trophoblasts, and thymic epithelial cells [95,96]. Interestingly, the trophoblasts do not express HLA-A and HLA-B but only HLA-C, HLA-E, HLA-G [96], and HLA-F [97]. The induction of HLA-G in macrophages and monocytes with proinflammatory cytokines such as IFN-γ was observed [98]. Gonen-Gress et al. [99,100] demonstrated the expression of Face-2 of HLA-G in a trophoblast cell line, Jeg-3, by in situ immunostaining.

In contrast to HLA-Ia molecules, the export of HLA-F from the ER requires the cytoplasmic tail, as observed in HeLa cells, possibly for a novel function, namely homodimerization (vide infra) [101]. The expression of HLA-F was first observed on the surface of B lymphocyte and monocyte cell lines and in vivo on extravillous trophoblasts that had invaded the maternal decidua [102]. Further studies revealed cell surface expression of Face-2 of HLA-F upon activation of B cells, T cells, NK cells, and monocytes using PMA and ionomycin [103]. Dulberger et al. [104] pointed out that there are ten conserved a.a residues that are critical for the architecture of the peptide-binding groove and for binding with specific peptide residues. Five of these ten residues are altered within the HLA-F groove, which led Geraghty et al. [105] to predict HLA-F either incapable of presenting peptides or doing so in a different manner from other HLA-I molecules. A number of investigations have also failed to elute canonical HLA-I peptides from natively expressed HLA-F from human cell lines, leading to the conclusion that HLA-F may not present antigen [106] but may act as a receptor for ligands to mediate signal transduction [107].

**Inference:** Face-2 of non-classical HLAs is also found on activated T and B cells, NK cells, monocytes, and placental extravillous cytotrophoblasts.

## 7. Does Face-2 of HLA B27 Alleles Play a Role in Spondyloarthropathies?

An association between HLA-B27 and spondylorthropathies was observed in the 1970s [108,109]. When Parham’s group [89,110,111] observed the presence of B2m-free HCs of HLA-B27 bound with influenza peptides on the surface of human HLA-B27-transfected B cell lines (HLA-B27-CIR) as early as 1991, they noted that the “peptides seems to bind empty (B2m-free) HLA-B27 molecules that are relatively stable under physiological conditions. The possibility that certain class I molecules (B2m-free HCs) present extracellular peptides to T cells in vivo should therefore be considered. For HLA-B27, this property might underlies the strong association of this allele with ankylosing spondylitis and other seronegative spondylarthropathies” (p. 77, [89]). Subsequently, many investigators have observed that HLA B27 alleles have a strong association with spondyloarthropathies [112,113,114,115,116]. Spontaneous arthritis was also observed in HLA-B27 transgenic mice lacking B2m (HLA-B27+ B2m−/−) but not in HLA-B27+ B2m+/− mice. To determine whether processing, assembly, transport, and expression of the HLA-B27 molecule play a role in the disease process, Khare et al. [117] introduced the HLA-B27 transgene into B2m-deficient mice. Within 2–4 weeks after transfection, most of the male mice possessed B2m-free B27 HCs (HLA-B27-Face-2), but not with B2m-associated HLA-B27 (B27-Face-1). The mice with B27-Face-2 spontaneously developed nail and joint changes, hair loss, and swelling in paws, which lead to ankyloses. Thirty-three of forty-for (75%) male mice 4 months or older developed arthritis compared to seven of twenty-three (30%) female mice. To determine whether the HLA-B27-Face-2 reached the cell surface in B2m−/− HLA-B27 mice, they stimulated the splenocytes in vitro with Con-A. Low-level expression of HLA-B27-Face-2 was detected on the cell surface of Con A-stimulated splenocytes. The presence of Face-2 on the cell surface after stimulation suggested that environmental factors can stimulate cell surface expression of Face-2. Furthermore, Khare et al. [118] demonstrated that Face-2 contributes to disease pathogenesis in B27 transgenic mice. Spontaneous disease was delayed by treatment of anti-HC-specific mAb, suggesting the importance of B27-Face-2 in disease pathogenesis. Interestingly, they suggested that the HLA-B27-Face-2 may function like an HLA class II molecule, possibly by forming dimers (Face-3/Face-4). This is significant in the evolving concepts on B2m-free HLA variants.

Vasquez et al. [119] studied two untransfected cell lines, Hmy2.CIR and a human lymphoid cell line T2, after transfecting with various HLA-B27 alleles (B*27:04, B*27:05, B*27:06, and B*27:09). Flow cytometric observations using an intact (Face-1) HLA-B27-specific mAb ME1 (IgG1) and an HC-specific mAb HC10 revealed that irreversible forms of Face-2 appeared at the cell surface to a similar extent among all subtypes, irrespective of their association with Ankylosing Spondylitis. Bird et al. [120] recorded that some of these lymphoblastoid cells expressed HLA-B*27:05 homodimers (Face-3) both intracellularly and at the cell surface following endosomal recycling.

Studying the white population in the UK, Brown et al. [121] pointed out that infection of HLA-B27-expressing cells by an organism may trigger spondyloarthropathy. Such infection may interfere with the cellular antigen-presenting function and contribute to the expression of aberrant HLA-B27 homodimers (Face-3). Orchard et al. [122] found that, in the peripheral arthropathies of inflammatory bowel disease, the cell-surface HLA-B27-homodimers may engage NK or related immunoreceptors expressed on lymphocytes or other cells within the joint, to elicit local cytokine production or enhanced cellular activity and to perpetuate joint inflammation. These observations indicate that HLA-I variants, not only Face-2 but also Face3 and Face-4, may play a role in the pathogenesis of spondyloarthrpathies. These variants may serve as biomarkers of disease.

Most importantly, Belaunzaran et al. [123] elucidated the functional role of HLA-B27-homodimers (Face-3 of HLA-B27) using a B27-homodimer-specific mAb (HD5). The mAb HD5 bound specifically to HLA-B27-homodimers but not B2m-associated intact HLA-B27 or A1, B7, B13, or C7. The HLA-B27-Face-3 binds to members of the killer cell receptor (KIR) and leukocyte Ig-like families (details vide infra). In addition, in vitro assays demonstrated that HLA-B27-Face-3 interactions with CD4+ T cells induced proinflammatory responses (secretion of TNF and IFN-γ). Therefore, it was postulated that mAb HD5 may block the interaction of HLA-B27-Face-3 to alter the immune responses. Indeed, mAb45 resulted in reduced soluble TNF and decreased CD4+ T cells. This investigation of HD5 mAb necessitates the need for developing therapeutic human HD5-specific monoclonal antibodies for treating spondyloarthropathies. Furthermore, investigating serum antibodies against HLA-B27-Face-3 in spondyloarthropathies may yield more information on the pathogenesis and progression of the disease and enable the development of other specific and personalized therapies.

**Inference:** In both mice and humans, HLA-B27-Face-2 and their dimers are associated with spondyloarthropathies. The anti-HC antibodies, which recognize Face-2 and their dimers, minimize spontaneous arthropathies.

## 8. Current Concepts Concerning Oligomerization of Face-2

### 8.1. Is the Cell Surface Clustering of Face-2 a Causal Factor for Dimerization to Generate Face-3 and Face-4 Variants?

It appears that dimerization is promoted by the clustering of Face-2 on the cell surface. Several reports document that the Face-2 molecules on the cell surface are capable of aggregating with other Face-2 molecules of HLA-I. In this regard, the observations of Chakrabarti et al. [124] are notable. While constituting liposomes, they observed the self-association or clustering of fluorescinated Face-2 of HLA-A2. The photographic figures presented in the report documented molecular proximity of the self-associated Face-2 molecules, as determined by flow cytometric phosphorescence resonance energy transfer (FCET). The figures also enable the visualization of the aggregation of the HCs monitored by fluorescence photo-bleaching recovery (FPR) and time-resolved phosphorescence anisotropy (TPA), in addition to FCET. The aggregation was blocked by B2m added to the liposomes. HLA aggregates, defined by FCET, were readily detected on the surface of human lymphoblastoid (JY) cells. Similarly, Matko et al. [125] detected clustered Face-2 molecules on activated normal B and T cells, on cells of B and T lymphoblast lines, and on transformed fibroblasts. No clustering was observed on the surfaces of resting B or T cells or normal fibroblasts. The Face-2 clustering correlated with the presence of the HC-10 epitope of Face-2 at the cell surface. The clustering was reversed by exogenous B2m. Several other investigators have confirmed the occurrence of HLA Face-2 clusters on the plasma membranes of human T (HUT-102B2) and B (JY) lymphoma cells [126]. The possible role of Face-2 oligomerization (formation of Face-3 and Face-4) was further studied on a B lymphoblastoid cell line, JY [127]. Although these unique reports established further progress in the evolving concept of different faces of B2m-free HCs of HLAs, further stringent evaluation of the different faces is required to further validate the hypothesis of Khare et al. [117] that the Face-3 and Face-4 variants can function like HLA class II molecules.

Using super-resolution microscopy, Kennedy et al. [128] examined variations, if any, in the cell surface organization of different alleles at the nanoscale level. HLA-I alleles (HLA-B*27:05/B*53:01/B*57:01 and HLA-C*06:02/C*07:02) were transfected into LCL. 721.221 lymphocytes that lacked the expression of classical HLA-I. After sorting cells to express HLA-I molecules at similar levels, they were placed on poly-L-lysine-coated glass, stained with mAb W6/32 that binds to intact HLA-I (Face-1) and imaged by stochastic optical reconstruction microscopy (STORM). This technique provided co-ordinates for the location of HLA-I, with a resolution of <20 nm. Allotypes of the same locus did not differ in their nanoscale organization. The clusters of HLA-C on the cell surface were larger than for HLA-B, revealing that different loci exhibit distinct nanoscale organizations at the cell surface. Although mAb W6/32 was used with the notion that it may stain only intact B2m-associated HLA molecules, work of Tran et al. [86] showed that the mAb W6/32 “actually recognizes an epitope present on isolated non-reduced α-chains of most HLA-B allelic forms”. These studies suggest that clustering of cell surface HLA molecules may have facilitated dimerization of B2m-free HCs.

**Inference:** Cell surface clustering of Face-2 promotes its oligomerization. Clustering and oligomerization of Face-2 is prevented by exogenous B2m, suggesting that B2m can downregulate the dimerization of Face-2. The half-life of Face-2 and its dimers requires detailed investigation.

### 8.2. Is Dimerization a Necessary Prerequisite for Face-2 on the Cell Surface?

The dimerization of HLA HCs was investigated on H-2L^d^and H-2D^b^ MHC class I molecules of the mouse by Capps and Zuniga [129,130]. They generated antiserum against a synthetic peptide corresponding to a portion of the cytoplasmic domains of the HCs of H-2L^d^and H-2D^b^. The antibody bound exclusively to B2m-free H-2L^d^/D^b^ HCs. Immunoblots of cell lysates were immunoprecipitated with R4-reactive H-2L^d^/D^b^ which revealed the presence of homodimers and heterodimers of B2m-free HCs. The B2m-free HC molecules arose late in their biosynthesis. Clustering of cells with Face-1 or with exogenous B2m molecules prevented the formation of H-2Ld/Db HC dimers. It was concluded that dimerization (formation of Face-3 and/or Face-4) could be a consequence of the loss or unavailability of B2m. In addition, the dimerization of Face-2 is interpreted as a strategy to eliminate dysfunctional monomers (Face-2). Similarly, Allen et al. [131] observed homodimerization of Face-2 on the cell surface HLA-B27-transfected T2 cells. Tran et al. [132] reported a correlation between disease incidence and HLA-B27 dimers in the HLA-B27 transgenic rat models.

Lynch et al. [133] demonstrated that a significant proportion of the HLA-I content of exosomes exists in the form of disulfide-linked dimers. These dimers can be detected after the release of exosomes from human monocyte-derived dendritic cells. They also found that heterodimers (Face-4) formed between two different HLA-I alleles. Most importantly, they have shown that exosomes in the human EBV-transformed B cell line (Jesthome) may form Face-4 between HLA-A and -B molecules. Similarly, Makhadiyeva et al. [134] showed that similar Face-4 dimers can be detected on the human lymphoblastoid lines LCL.221, CEM, Jesthome cell line, and on the rat C58 lymphoma cell lines when the redox environment has been significantly altered, either by chemical oxidation with diamide, chemically induced apoptosis with hydrogen peroxide and thimersol, or by cross-linking of FacR/CD95. Interestingly, in the CEM cell lines, there were low levels of HLA-B27 dimers in the cell lysates in the absence of oxidative stress, whereas the Jesthome cell line, which expresses a higher level of HLA-B27 than the CEM cell lines, displayed dimers under normal conditions. These findings support the contention of Santos et al. [135] that Face-3 and Face-4 dimers in dendritic cells form only after the activation induced upregulation of Face-2. Morales et al. [136] observed the dimerization of B2m-free HLA-G (Face-3) in the villous cytotrophblasts. HLA-F is capable of homodimerizing (Face-3) with the Face-2 of other HLA-F molecules or heterodimerizing (Face-4) with other HLA-I loci, such as HLA-C [101,137].

There is an urgent need to investigate the possible occurrence of diverse forms of homo- and heterodimers involving the six haplotypes of HLAs. We envisage diversified forms of homo- and heterodimers and diagrammatically illustrated the hypothesis in Figure 4, for they may have significant functional and immunological implications in various disease states including tumorigenesis, metastasis, end stage organ diseases, autoimmunity, organ transplantation, or allograft tolerance and rejection. We hypothesize that an individual who has identical pairs of alleles of all six isomers is capable of forming up to 36 different kinds of homo- and heterodimers and, in an individual who has two different pairs of alleles of all the six isomers, 144 different kinds of homo- and heterodimers may be possible. Dimerization may occur in different alleles at different time points depending on the nature of the activating factor, implying that the dimerization may differ under different pathological conditions. The functional potential of these homo- and heterodimers requires extensive experimental investigation, further developing the work conducted by previous investigators [129,130,131,132,133,134,135,136,137].

Not only should HLA typing be known of patients for developing personalized immunotherapy, but a consideration of the immunogenicity and antigenicity of homo- and heterodimers is also potentially critical. Based on Figure 4, we visualize approximately 144 monoclonal antibodies specific for each one of these dimers. Similarly, the patients’ sera have to be screened against these monomers and dimers and most importantly against the dimers because they interact with both KIR and the Leukocyte Ig-like family of receptors.

We hope that our hypothesis may stimulate further studies on the multivarious HLA-I antibodies formed against novel HLA-I heterodimers in mismatched organ transplantation, which may play a role in transplant rejection. Indeed, proinflammatory cytokines and T cell activation can be expected after recognition of donor HLA antigens, either by the direct pathway (donor HLA on donor antigen-presenting cells), the semidirect pathway (intact donor HLA on recipient’s antigen-presenting cells), or by the indirect pathway (self-restricted presentation of processed donor-derived HLA HCs on HLA molecules on recipient antigen-presenting cells) [138]. In this regard, Horuzko’s team of investigators [139,140,141,142] have observed HLA-G with a high level of HLA-G dimers in circulation and augmented expression of the membrane-bound HLA-G on monocytes which are associated with the prolongation of kidney allograft survival. Interestingly, the HLA-G dimer in circulation was higher in a group of 90 patients with a functioning allograft compared with 40 patients who underwent allograft rejection. They have demonstrated that the HLA-G dimer inhibits the activation and cytotoxic capabilities of human CD8+ T cells. This mechanism is implicated in the downregulation of Granzyme B expression. Such homo- and heterodimers may function through differential binding to the LILRBI family of receptors [141]. They also documented the presence of soluble HLA-G dimers as being associated with lower levels of proinflammatory cytokines, suggesting a major role of HLA-G dimers in controlling an inflammatory state.

Similarly, the possible role of these dimers in the lymph node and organ metastasis of human cancer as well as in autoimmune diseases needs to be investigated for developing personalized active specific and passive immunotherapies. All these observations suggest that Face-2 may be transitory HLA molecules generated under activation or inflammation-induced oxidative stress, and Face-2 may be immediately transformed into a homodimer (Face-3) or heterodimer (Face-4) depending on the nature of the alleles in the vicinity. Since Face-2 exposes shared and unique sequences (^115^QFAYDGKDY^123^ and ^137^DTAAQI^142^) due to the absence of B2m, it may elicit potentially detrimental antibodies. The formation of Face-3 and Face-4 may serve a preventative function against possible ill effects of binding of such antibodies to exposed epitopes on Face-2.

Possibly, these homo- and heterodimers of HLA-I Face-2 may also represent neo-HLA-I molecules or classes, from the point of view of the genes responsible for the HCs of HLA classes, as illustrated in Figure 5. More structural studies are needed to understand their possible functions and role in human pathology. Incidentally, it is interesting to note that Triantafilou et al. [143] observed the oligomerizaton of HLA class II in HLA-DR-transfected fibroblast cells by immunoprecipitation techniques from detergent-derived cell extracts.

**Inference:** Although dimerization (formation of Face-3 and/or Face-4) could simply be a consequence of the loss or unavailability of B2m, dimerization masks the exposure of the most commonly shared highly immunogenic amino acid sequences in the HC, which are masked by B2m in Face-1. Therefore, dimerization could be a strategy to protect highly immunogenic sequences on the Face-2 molecules.

### 8.3. Do Cysteine Residues in Face-2 Facilitate and Stabilize Dimerization?

Allen et al. [131] showed that HLA-B27 Face-2 can form disulfide-bonded homodimers (Face-3) through cysteine residue at position 67 (C^67^) in the extracellular α1 domain. It is reasonable to infer the role of C^67^ in HLA-B27 homodimerization, since 41 of the 44 HLA-B27 alleles possess C^67^ (Table 3), with the exception of B*2718, B*2723, and B*2729. In contrast to HLA-B27 Face-2, B27 Face-3 homodimers appear to be binding to peptides and stabilized by a peptide epitope, despite the absence of B2m. Most interestingly, HLA-B27 Face-3 homodimers are still recognized by both Face-2 binding mAb HC10 and Face-1 binding mAb W6/32. Although Allen et al. [131] believed that mAb W6/32 is conformation-specific, surface labeling and immunoprecipitation showed the presence of mAb W6/32-reactive Face-2 at the surface of HLA-B27-transfected T2 cells. McMichael and Bowness [115] observed that disulfide-bonded HLA-B27-Face-3 is present in HLA-B*2705-transfected LBL721.220 cells. mAb HC10 confirmed the presence of both non-reduced and reduced Face-3 in Western blots. In this regard, Mear et al. [144] proposed that the Face-3 homodimerization is a result of HLA-B27 “misfolding” within the ER, the accumulation of which occurs concomitantly with the proinflammatory intracellular stress response. Such misfolding of HLA-B27 may contribute to its recognition by both HC10 and W6/32.

Antoniou et al. [145] studied the formation of B27-Face-3 and the relationship to assembly kinetics. Two distinct populations of HC-dimers were detected, one within the ER and another at the cell surface [120,145]. One set of homodimers form in the presence of C^67^ [146] and another in the absence of C^67^ [120]. Evidently, cysteine residues other than C^67^ or other amino acids, such as serine or tyrosine, could be involved in HC dimerization. Taurog [147] reported that transgenic rats expressing HLA-B27 with serine substituted for C^67^ developed the spondyloarthropathy-like phenotype with somewhat less arthritis than controls expressing wild type B27, suggesting that C^67^ may not be a prerequisite for the disease. Importantly, C^67^ is also shared by a small cohort of other HLA-B alleles (Table 3).

Bird et al. (120) have shown “that presence of C^67^ alone in B15 and B39 HCs is not sufficient to form homodimers” (p750), although the authors failed to note that there are some B39 alleles without C^67^ (see Table 3). Indeed, HLA-B39 is also reported to be associated with spondyloarthropathy [148,149]. B27 mutants with C^67^ can generate S-S bonds with intra-chain cysteines of the α2 and α3 domains; however, these misfolded forms may not exit the ER in human cells (114). Similarly, Antoniou et al. [145] suggest the possibility of HC dimerization through bonding between C^67^ of one HC with C^164^ of another HC. It is evident from this report that B2m-free HCs within the ER can form dimers through the structurally conserved Cys at position 164 (Table 3). Nossner and Parham [150] also observed the dimerization of HLA-B*07:02-transfected C1R cells, which is an HLA-A- and HLA-B-deficient EBV transformed B cell line expressing HLA-C4.

All these observations suggest that dimerization could result not only in homodimers (Face-3) but also in heterodimers (Face-4). Antoniou et al. [145] further pointed out that residues surrounding C^67^ can promote HC dimerization. In this regard, Whelan and Archer [151] have shown Lys^70^ residue as observed in HLA- B27 and B73, conferring enhanced chemical reactivity to C^67^. Antoniou et al. [145] demonstrated that Lys^70^ and Ala^71^can impact dimerization when substituted into HLA-A2. They point out that residues Gln^65^ and Iln^66^ can also enhance HC dimerization. If this is verified, it may further confirm that heterodimerization can occur among different alleles of different haplotypes.

**Inference:** Dimerization of B2m-free HCs may involve cysteine–cysteine linkages between two heavy chains. In HLA- B-derived HCs, C^67^ is highly prevalent and may contribute to S-S bonding between the HCs. Other amino acids adjacent to C^67^ such as Gln^65^ Iln^66^ Lys^70^ Ala^71^can impact S-S linkages of the dimers.

### 8.4. Documentation of Cysteine-42-Mediated Homodimerization of HLA-G

Boyson, in Strominger’s group [152], investigated whether HLA-G dimerization occurred naturally, using the HLA-G-transfected mutant EBV-transformed B cell line B-LCL 721.221 which is devoid of any HLA class I molecules. The B-LCLs were lysed and analyzed on SDS/PAGE gels under both reducing and non-reducing conditions, blotted to nitrocellulose. On the blots, the HLA-G HCs were monitored with the HLA-G HC-specific mAb MEM-G/1. Under non-reducing conditions, two bands, one 39 kDa and another 78 kDa, were observed. However, under reducing conditions, the 78 kDa band was not observed, suggesting that the band missing in the reducing gels was a dimer, possibly disulfide-bonded. Further confirmation of the dimeric nature of the 78 kDa band was ascertained by a parallel control experiment. They repeated the experiments conducted by another Strominger team [153] in which HLA-A2 was immunoprecipitated by using anti-β2m mAb (BBM.1) beads from surface-biotinylated HLA-A2 transfectants in the presence and absence of iodoacetamide (IAM). Dimerized HLA-A2 HCs were observed under nonreducing conditions in the absence of IAM, but this dimerization was completely abrogated by the addition of IAM, indicating that the HLA-A2 dimers could be an artifact of cell lysis and immunoprecipitation. The inclusion of IAM in the lysis buffer enabled distinguishing between preexisting and artefactual HLA dimers. Similarly, when HLA-G was immunoprecipitated from surface-biotinylated HLA-G transfectants, HLA-G dimers were detected under nonreducing conditions even in the presence of IAM, suggesting that they were preexisting dimers and not an artifact. The existence of dimers was confirmed using the conformation-specific W6/32 mAb and the HLA-G-specific MEM-G/11 mAb. Furthermore, the mutation of C^42^ on HLA-G to Ser^42^ completely abrogated the dimerization of HLA-G in the absence of IAM. Thus, dimerized HLA-G exists on the cell surface linked by means of a C^42^-mediated disulfide bond. Gonen-Gross et al. [99,100] documented the involvement of C^42^ in the dimerization of HLA-G using the B cell line LCL 721.221 and a melanoma cell line LB33 mel B1 that expresses HLA-A24 only. Similarly, Shiroishi et al. [154] showed that HLA-G can be expressed as a disulfide-linked dimer both in solution and at the cell surface. Although Apps et al. [155] contend that HLA-G is a B2m-associated dimer with increased avidity for LILRB1 receptors, Morales et al. [136] showed that normal placental villous cytotrophoblast cells indeed synthesized B2m-free, S-S-bonded HLA-G HC homodimers. Interestingly, incubating cells at reduced temperatures enhances HC dimerization in both HLA-B27 and HLA-A2 [145]. It is interesting to note that HLA-A2 (A*02:07, A*02:15N, A*02:18) has C^99^ (Table 4). Whereas Cys is lacking at position 67 in HLA-A, HLA-C, HLA-E, HLA-F, and HLA-G, it occurs at positions 101, 164, and 203 in HLA-A (Table 4), and in HLA-C, it occurs in these positions as well as in the terminal position # 1 (Table 5).

**Inference:** During dimerization, C^42^ may be involved in S-S linkage in B2m-free HCs of HLA-G. The finding that the mutation of C^42^ on the HCs to Ser-42 totally abrogated dimerization confirms the role of C^42^ in the dimerization of Face-2.

### 8.5. Does Oxidative Stress Play a Role in Dimerization?

Davies and his team [156,157,158,159,160,161] have extensively studied the role of covalent crosslinking within or between polypeptide HCs of different proteins. They point out that such covalent crosslinking within and between HCs plays a role in determining structural configurations and ligand-receptor interactions of the proteins. Under normal physiological conditions, covalent crosslinking can occur either enzymatically or due to molecular reactions. Under pathological conditions (inflammation, injury, infections, and malignancy), “the covalent crosslinking is generated as a consequence of exposure to oxidants (radicals, excited states, or two-electron species) that are induced by endo- or exogenous stimuli and as a result of the actions of a number of enzymes (oxidases and peroxidases)” (p. 1, [160]). The oxidative crosslinking may result in the folding of HCs within cells in the ER or Golgi involving (1) the generation of S-S bonds between two Cys residues, and (2) crosslinking of two Tyr residues (Figure 6). These crosslinkings occur commonly during oxidative stress. They can be formed between different sites within the same molecule (intramolecular or intrachain crosslinks) or between two different chains in a single molecule (e.g., the interchain crosslinks in mammalian insulin receptors), and they play a key role in stabilizing or maintaining the structural configurations that are essential to functional activity. The number of S-S bond-containing polypeptide HCs is well defined in structural proteins such as receptors (e.g., the low-density lipoprotein receptor, LDLR), and extracellular matrix proteins (e.g., laminin). Similarly, dityrosine crosslinking is observed in lipoproteins, extracellular matrix proteins, lysozyme, myoglobin, fibronectin, laminins, calmodulin, insulin receptors, hemoglobin, and centrin 2 [161]. Dityrosine crosslinking may arise due to oxidative damage in cells exposed under stress or exposure to continuous flux of H_2_O_2_ or exposed to peroxidase activity. However, persistent inflammation and oxidative stress are well known in patients with end-stage renal disease (ESRD) [162,163,164,165,166].

Interestingly, Colombo et al. [166] performed a detailed investigation on plasma protein-bound di-tyrosines as biomarkers of oxidative stress in ESRD patients on maintenance hemodialysis (HD). They observed that the pre-HD levels of plasma protein-bound di-tyrosines in hemodialyzed patients were significantly higher when compared with di-Tyr levels in age-matched healthy subjects. “In most ESRD patients, a single HD session decreased significantly the plasma protein-bound di-Tyr level, even if the mean level of plasma protein-bound di-Tyr post-HD remained significantly greater in ESRD patients compared to the mean di-Tyr level in age-matched healthy subjects” (p. 60, [153]). Also, urine contains polypeptides with dityrosine in patients with diabetes undergoing HD [166]. Evidently, inflammation-induced oxidative stress to kidneys in ESRD may have caused the release of proteins containing di-tyrosine into circulation. However, the exact source of the dityrosine-containing polypeptides in circulation remains uncertain and needs to be investigated. Interestingly, there is no report to date documenting the dimerization of Tyr or dityrosine of HLA HCs, although (1) dityrosine is observed in plasma and urine from patients with diabetes, ESRD, or undergoing hemodialysis [166], and (2) Tyr is abundant in HLA-A (Table 5) and in isomers of HLA-Ib.

**Inference:** During inflammation, injury, infections, malignancy, ESRD, and transplantation, covalent crosslinking may be generated by oxidative stress induced by endo- or exogenous stimuli and because of the actions of a number of enzymes (oxidases and peroxidases). The oxidative crosslinking may result in the folding of HCs within cells in the ER or Golgi involving S-S bonds between two Cys residues, and/or crosslinking between two Tyr residues.

## 9. Diversified Functional Capabilities of B2m-Free HLA Variants

One of the primary roles of B2m in intact HLA class I (Face-1) is to provide a stable groove for the binding of peptides for antigen presentation to the CD8+ T lymphocytes. HLA-I (Face-1) may also play a role in non-immunological functions by interacting with insulin receptors, endorphin, glucagon, and epidermal growth factors [167,168,169,170,171]. Immunological and non-immunological functions of B2m-free HLA variants are steadily emerging. The above reports claim that peptides can bind to these variants. In a transfected B cell line (LCL 721.221), the peptides bind to glycosylation-altered Face-2 of HLA-F [103]. The peptides can bind to HC in the absence of B2m “with sufficient affinity to establish a functionally conformed molecule on the cell surface (p. 833, [17])”. Carreno and Hansen [94] confidently claim that the binding of exogenous peptide ligand can influence the expression and half-life of B2m-free HLA variants at the cell surface. However, there is no report so far to document that the HLA variants present antigens.

Recent findings indicate that the α1 and α2 domains of HLA variants can serve as binding sites for short peptides, polypeptides, and a variety of proteins. Like HLA-I Face-1, B2m-free HC variants (Face-2, Face-3, and Face-4) (Figure 3) have the propensity to bind to members of killer cell receptor (KIR) and leukocyte Ig-like families. Kulkarni et al. [172] clarified that the interaction between KIR and HLA-I results in the regulation of immune responses in infection, inflammation, and malignancy. Detailed investigations have been carried out by Kollnberger and his team of investigators [173,174,175,176,177,178,179,180,181,182,183] on KIR and HLA-I variants in spondyloarthropathies and other diseases. KIR receptors are categorized by the number of extracellular Ig-like domains, either 2 (termed 2D) or 3 (termed 3D). KIR3D includes domain 0 (D0), D1, and D2, whereas KIR2D may have D0 and D2 or D1 and D2. KIRD may have long (KIRDL) or short (KIRDS) cytoplasmic tails and they perform a variety of roles in immune homeostasis. KIRDL performs inhibitory immune regulatory functions, whereas KIRDS performs immune stimulatory functions. Their domains interact differently with different B2m-free HLA variants and HLA haplotypes [173,175,176,177,179,180,181,182,183]. KIR3DL2 expressed on NK and T cells binds to B27-dimers of Face-2, and Face-2 per se on antigen presenting cells, tumor cells, and trophoblast cells. D0 plays a central role in the stronger binding of KIR3DL2 to Face-2 of HLA-F, and Face-2 and Face-4 of HLA-I classes.

Kollnberger’s team [173,174,175,176,177,178,179,180,181,182,183] assessed KIR binding to HLA-Ia variants using several approaches, which included (i) flow cytometric analysis of KIR-expressing cell lines stained with HLA-I Face-1; (ii) recombinant KIR Fc fusion proteins or KIR tetramers coated fluorescent multiplex beads; (iii) physical measurements of KIR binding to HLA-I variants; (iv) binding of KIR-expressing reporter cells to HLA-I variants or to HLA-I variant transfected cell lines; and (v) KIR inhibition of NK cytotoxicity and IFN-γ production by binding to HLA-I variants. The *binding of HLA variants inhibits lymphocyte IFNγ production and cytotoxicity. KIR3DL2 has the propensity to suppress tumor immune surveillance and promote maternal fetal immune tolerance.* Since KIR family receptors are not restricted to subsets of NK cells but are also expressed by γδ, CD8 and CD4 αβ T cells involved in health and various diseases [184,185], more detailed investigations are needed to assess the outcome of interactions of KIR-HLA variants on these different kinds of T cells.

Interestingly, when immune cells are activated due to inflammation, infection, injury, and pregnancy, the B2m-free HLA variants (Face-2, 3, and 4) such as that of HLA-C and HLA-F are overexpressed [186,187,188,189]. Concomitantly, KIRs are also activated and consequently the HLA-I variants act as ligands for activating KIRs. During pregnancy, the secretion of cytokines and growth factors that enable vascularization are essential for blood supply through the placenta [190,191,192]. As a result, NK cells in the endometrial decidual tissue, both KIR2DL1/2, as well as monomeric, homo- and heterodimeric variants of B2m-free HLA-F, HLA-C, and other HLA-I are activated for functional interactions [177,179]. The variants of HLA-F and other HLA-I loci act as ligands for activating KIR and KIR3DL2 expressed on placental and maternal cells [192], suggesting that not only the suppression of inhibitory function but also the activation through KIRs may contribute to normal pregnancy. A complementary report [193] suggests that HLA-F variants act as ligands for KIR3DS1. However, more focused experimentation on activated cells under different conditions such as inflammation, infection, transplantation, spondyloarthropathies, malignancy, and metastasis are needed for deriving definitive conclusions on the interaction between activating and inhibiting KIRs and specific HLA-I variants.

**Inference:** The expression and half-life of B2m-free HLA variants at the cell surface may depend on ligands binding to the variants. The ligands may include exogenous peptides, members of the KIR family with long and short cytoplasmic tails, and leukocyte Ig-like molecules, which are expressed on several immune cells, including a variety of T cells and NK cells.

## 10. Are B2m-Free HC Variants the Evolutionary Progenitors of HLA-I and II and May Represent Proto-HLA?

The structures and functions concerning HLA-I and HLA-II are found throughout jawed vertebrates. Extensive studies carried out by Dijkstra and co-investigators [194,195,196,197,198,199] on their broad distribution and conservation during vertebrate evolution demonstrated that the structural patterns of HCs are evolutionarily primitive among HLA-I and HLA-II classes. Structural configuration of HLA-I has been studied in cartilaginous and bony fishes, amphibians, birds, and a variety of mammals, whereas HLA-II structures were studied in lungfish, chicken, mice, and humans (for detailed citations, see Wu et al. [1,199]). Comparing the sequences of HCs, Kaufman and co-investigators [200,201,202] postulated that B2m-free HC-homodimers would have evolved into class II heterodimers, which would have given rise to ancestral HLA-I and HLA-II. A similar postulation was made by Hashimoto and others [203,204,205] on the origin of HLA classes. Concurring with this postulation, Wu et al. [1] have recently examined in detail the structure and sequences of HLA-I and HLA-II from the perspective of their relationship, and they developed a diagrammatic model with which the authors proposed in the title itself that “in evolution, a class II-like molecule came first”. Although it is thought-provoking and interesting, hitherto no specific investigation has been undertaken to examine the presence of B2m-free HCs of HLA molecules, primarily due to the contention of earlier investigators that B2m-free HCs are unstable and that B2m provides stability to the HC. Even recently, while trying to explain how the synergistic binding of B2m, HC, and peptides, Wu et al. [199] state that B2m is stable when alone, while HC on its own is unstable. However, in vitro studies carried out recently by Dirscherl et al. [206], with isolated HCs in conjunction with molecular docking and dynamics simulations, suggest that the α3 domain of one monomer can dimerize with another monomer in a manner similar to that seen for B2m.

Hashimoto and others [203] have elegantly pointed out, based on the sequence similarities, that HLA class I and class II molecules are “most likely evolved from a common ancestor… The class II β-chain seems to have diverged more slowly than other chains… more than 400 million years ago the MHC class I-like molecules had membrane-proximal domains of the same length as the contemporary” (p. 2212) HCs of class II.

The reports recapitulated in the present review postulate that the B2m-free HLA–HCs of monomeric, homo-, and heterodimeric variants could be the common ancestors of classical and non-classical HLA-I and HLA-II. As a preliminary measure, we examined a.a sequence similarities between B2m-free HC (Face-2) and the heavy chains of HLA class II. Notably, the HLA-I sequence ^117^AYDGKDY^123^ is shared by classical (HLA-A, HLA-B, and HLA-C) and non-classical (HLA-E, HLA-F, and HLA-G) HLA-I and by Face-2 involved in homo and heterodimerization, although it is lacking in the HCs of HLA-II. However, we observed that another six a.a sequence (^34^ VRFDSD^39^), a neighboring doublet (RA), and a neighboring triplet (^58^EYW^60^) a.a. that are found in all classical HLA class I isomers are also found in the β-chain of DRB, DQB, and DPB isomers of HLA class II, as illustrated in Table 6.

Based on these preliminary observations, we have postulated an evolutionary tree for HLA-I and HLA-II with B2m- free HC as the progenitor, as shown in Figure 7. The conclusive proof lies in further scrutiny of the a.a. sequences and in documenting the presence of B2m-free HCs from cephalochordates and Agarthans, although in these two groups MHC genes are reported to be lacking [1]. The polyreactive mAbs such as TFL-006 or TFL-007 and other mAbs as listed in a recent report [37] likely aid in the diagnosis of Face-2 upon activation of cells, particularly on the cell surface of immune cells of Agarthans, cartilaginous fishes, and higher vertebrates, with proinflammatory cytokines, chemokines, PHA, PMA, and such similar activators.

**Inference:** The presence of six a.a. sequences (^34^ VRFDSD^39^) with a neighboring doublet and a triplet (RA and ^58^EYW^60^), which are found in all classical HLA class I isomers, are also found in the β-chain of DRB, DQB, and DPB isomers of HLA class II, suggesting that the different Faces of HLA variants, more particularly Face-2, may represent the proto-HLA and the progenitors of HLA-I and HLA-II classes.

## 11. Summary: Genesis, Transformation, and Functions of HLA Variants (Neo-HLA)

Since finding HLA-B8 and HLA-B27 B2m-free HCs (Face-2) on the cell surface of a human lymphoblastoid cell line [22], their genesis has remained controversial. However, the experimental investigations of Demaria et al. [11,12,13] clarified that Face-2 may emanate from two distinct pathways:(a)Inducing the expression of B2m-free HCs with PMA on activated T cells and blocking their exit with BFA from the ER together confirms that the induction of Face-2 on activated cells requires the direct transport via ER independent of their association with B2m.(b)In a second experiment, they stripped Face-2 from PMA-activated T cells by brief treatment with trypsin in the cold. The treatment of these cells with trypsin selectively removed surface B2m-free HCs but had no effect on B2m-associated HLA (Face-1) (p. 107, Lines 19–21, [11]). The treated cells were then incubated at 37 °C in the presence or absence of BFA. They noted that the BFA did not block Face-2 reappearance on these trypsin-treated activated T cells, indicating that they are dissociated from B2m-associated HLAs. They immunostained with mAb HC10 to confirm that the emergence of Face-2 is due to dissociation of B2m from Face-1.

These two observations put an end to the belief of many investigators that Face-2 results solely from the dissociation of B2m from Face-1. Furthermore, several reports have emerged postulating independent surface expression of Face-2, without dissociating from Face-1 [33]. One notable feature of Face-2 that emerged independently on the cell surface is a totally altered glycosylation pattern from that of Face-1, being either the absence of sialyl residues [90], differences in the density of mannose residues [22], or both.

HC-specific mAbs enabled the identification of the expression of Face-2 on the cell surface of activated T lymphocytes [33,84,85,123]. A causal factor for the emergence of cell surface expression of Face-2 is inflammatory stress, since T-cell activation occurs under the influence of proinflammatory cytokines. PHA, anti-CD3 mAb, and PMA result in the cell surface expression of Face-2 [11,84,85]. Increases from 25K to 60K molecules per T-cell after PHA stimulation [76] and increases on average from 30 to 65% after anti-CD3 mAb are indeed significant findings on the expression of Face-2 [85]. Indeed, low levels of expression of HLA-C and HLA-B Face-2 molecules are observed in normal human tissues such as normal epidermis, breast, lung, bronchi, esophagus, stomach, illum, colorectum, gall bladder, urinary bladder, seminal vesicles, ovarian epithelia [33,89], PBL, and spleen cells [91,92,93,94,95].

The half-life of B2m-free HCs (Face-2) seems to depend on peptide binding to the HCs [94] or on clustering and dimerization with other Face-2 molecules. While constituting liposomes, it was noted that Face-2 self-associated with other Face-2 molecules [121]. Similarly, Face-2 molecules cluster on activated normal B and T cells, on cells of B and T lymphoblast lines, and on transformed fibroblasts [122,123,124,125,126]. No such clustering was observed on the surfaces of resting B or T cells or normal fibroblasts. Interestingly, the clustering was reversed by exogenous B2m. Novel homo- (Face-3) and hetero dimers (Face-4) of Face-2 molecules are observed on the exosomes released from human monocyte-derived dendritic cells [131] and on different cell lines [132,133]. Dimerization of Face-2 molecules of HLA-G [134] and HLA-F [135] is also observed.

Under pathological conditions (inflammation, injury, infection, and malignancy), the dimerization may involve covalent crosslinking because of exposure to oxidants (radicals, excited states, or two-electron species) that are induced by endo- or exogenous stimuli and as a result of the actions of a number of enzymes (oxidases and peroxidases). The crosslinking can form between different sites within the same molecule (intramolecular or intrachain crosslinks) or between two different HCs and play a key role in stabilizing or maintaining the structural configurations that are essential to functional activity. The oxidative crosslinking may result in folding of HCs within cells in the ER or Golgi involving (1) the generation of S-S bonds between two Cys residues, and (2) crosslinking of two Tyr residues (Figure 6).

Studies carried out on spondyloarthropathy suggest that the Face-2 dimerization is caused by Cysteine^67^ residues in the α-domain of the HCs of HLA-B alleles [144,145,146,147,148,149,150,151]. Spondyloarthropathy is highly minimized when C^67^ is replaced by serine in transgenic rats [147]. One set of homodimers forms in the presence of C^67^ [134] and another in the absence of C^67^, suggesting the possible involvement of Cys in positions other than 67 [145]. C^42^ is involved in the dimerization of HLA-G [99,100,152]. Altering C^42^ to S^42^ completely abrogates the dimerization of HLA-G. Normal placental villous cytotrophoblast cells synthesize S-S bonded HLA-G homodimers (Face-3). Cells grown at reduced temperature seem to upregulate HC dimers in HLA-B27 and HLA-A2 [145]. Interestingly, HLA-A2 (A*02:02, A*02:18) has Cys at position 99. Table 3, Table 4 and Table 5 document the presence of tyrosyl residues in α1 > α2 > α3 domains of HCs. Di-tyrosine crosslinking is well documented in several proteins (lipoproteins, extracellular matrix proteins, lysozyme, myoglobin, fibronectin, laminin, calmodulin, insulin-receptors, hemoglobin, and centrin 2 [159,163,164,165,166]). It is yet to be investigated in the dimerization of HLA monomers, although dityrosine crosslinking is observed in pre-hemodialysis (HD) levels of plasma proteins. Di-tyrosines containing plasma proteins in hemodialyzed ESRD patients are significantly higher compared with levels in age-matched healthy subjects [166].

The primary factor promoting the dimerization of the monomers as well as the generation of covalent crosslinking under pathological conditions (inflammation, injury, infection, and malignancy) is oxidative stress induced by radicals, excited states, or two-electron species. Davies’s team [156,157,158,159,160,161] has clarified that the oxidative crosslinking may result in the folding of HCs within the ER or Golgi involving the generation of S-S bonds or even dityrosyl linkages to promote the dimerization of Face-2. Evidently, inflammation-induced oxidative stress to tissues may have released proteins containing di-tyrosine into circulation.

The possible functions of HLA HC homo- and heterodimers deserve more attention. B2m-associated HLA is well known to perform peptide presentation to CD8+ T cells. Since Demaria et al. [12,13] showed that the Face-2 molecules can remain unfolded, there is doubt regarding whether the monomeric HCs can bind to peptides. However, Giocomini and others [33] observed that approximately two-thirds of cell-surface-expressed free HLA-C1 HCs originate without B2m, and not due to dissociation from pre-formed trimeric complexes, and noted that “peptides can bind to HC in the absence of B2m with sufficient affinity to establish a functionally conformed molecule on the cell surface (p. 833)”. Such functionally conformed molecules on the cell surface of several cancer cells correlated well with the loss of expression of intact HLA-I (Table 2).

B2m-free HC HLA variant molecules may bind to peptides [14,15,16,17,87,94,103,129]. Most importantly, they may serve as ligands for different polypeptides and proteins, which include insulin receptors [6] and T cell receptor (α/β) molecules [29,30] as well as KIR and Leukocyte Ig-like families, expressed on NK cells, different subsets of T cells, and other immune cells. Interestingly, when immune cells are activated due to inflammation, infection, injury, or pregnancy, under the influence of cytokines and chemokines, not only are B2m-free HC HLA variant molecules (Faces 2–4) upregulated but different KIR molecules are as well. However, more in-depth research is needed to distinguish the functional consequences of the interactions between different allelic and isomeric, monomeric, homo, and heterodimers of B2m-free HLA variants, and different immune cell receptors.

Striking similarities between the a.a sequences (e.g., ^34^VRFDSD^39^ and ^117^AYDGKDY^125^) located on the independently expressed monomeric B2m-free HLA variants of HLA class I and class II (^34^VRFDSD^39^) prompted us to explore the literature on the phylogenetic relationship among the two major HLA classes, in the context of independent expression of B2m-free HLA variants. We hypothesize that the monomeric B2m-free HLA-I could be the progenitor of not only homo- and heterodimeric variants but also HLA class I and HLA class II β chains. More investigations need to be conducted on the expression of polypeptides resembling monomeric and dimeric variants of B2m-free HLA in Agnathans and jawed cartilaginous fishes and on differences in the a.a sequences of α and β chains of HLA class II in higher vertebrates.

We hypothesize that the mono and dimeric variants may form a neo-HLA class. Figure 5 illustrates how the genes for the HCs of DR, DP, and DQ generate HLA class II. Similarly, the genes for the HCs of HLA-A, -B, -C, -E, -F, and -G have generated dimers without B2m. Therefore, there are three groups of HLA-gene products:(i)HLA–HCs associated with B2m (HLA-I);(ii)HLA–HC dimers belonging to HLA-II (DP/DQ/DR);(iii)The monomers, homo-, and hetero-dimers of HLA–HCs (HLA-A/-B/-C/-E/-F/-G).

The HLA-HCs of the third group are not formed by the dissociation of B2m but they are transported directly from ER and expressed independently of HLA class I [11,12]. Their α1 domain may remain unfolded. Their glycosylation patterns may differ from that of HLA-I. Above all, the structural configuration of the third group is strikingly different from that of HLA-I. In spite of the structural difference between Face-2 molecules, the α3 domain of one monomer can dimerize with another monomer in a manner similar to the dimerization of HCs with B2m [206]. There is no clear-cut evidence so far to document that the third-group HC dimers can perform antigen presentation. However, they can interact with unique receptors such as KIR molecules to upregulate and downregulate immune functions associated with NK, T, and B cells. Importantly, the third group is upregulated in cancer cells when group (i) is downregulated. Evidently, group (iii) may represent a neo-HLA class which may be the progenitors of other HLA classes of proto-HLA.

In conclusion, this review illustrates how different variants of HLA molecules emerge on the cell surface, independent of the B2m-associated HLA that we have designated as Face-1. Experimental investigations have demonstrated that variants (i.e., B2m-free monomers that we have designated as Face-2) can traverse from the ER to the cell surface independent of Face-1, particularly during activation of immune cells and other cells under the influence of proinflammatory cytokines and chemokines. The B2m-free monomers give rise to homodimers (Face-3) and heterodimers (Face-4) on the cell surface. Although we have speculated a variety of dimers based on Face-4, validation with further investigation is warranted. New reports are emerging on the functional diversity of the dimers in association with other cell surface receptors such as KIR and the leukocyte Ig-like family of receptors. These diversified functions in normal cells and malignant cells merit extensive investigation, which may shed light on organ metastasis. Similarly, we have postulated that the B2m-free monomer could be the phylogenetic progenitor of HLA-Class I and class II, as well as of the B2m-free dimers. Only focused future investigations may either confirm or reject these postulated hypotheses.

## Figures and Tables

**Figure 1 biomolecules-13-01178-f001:**
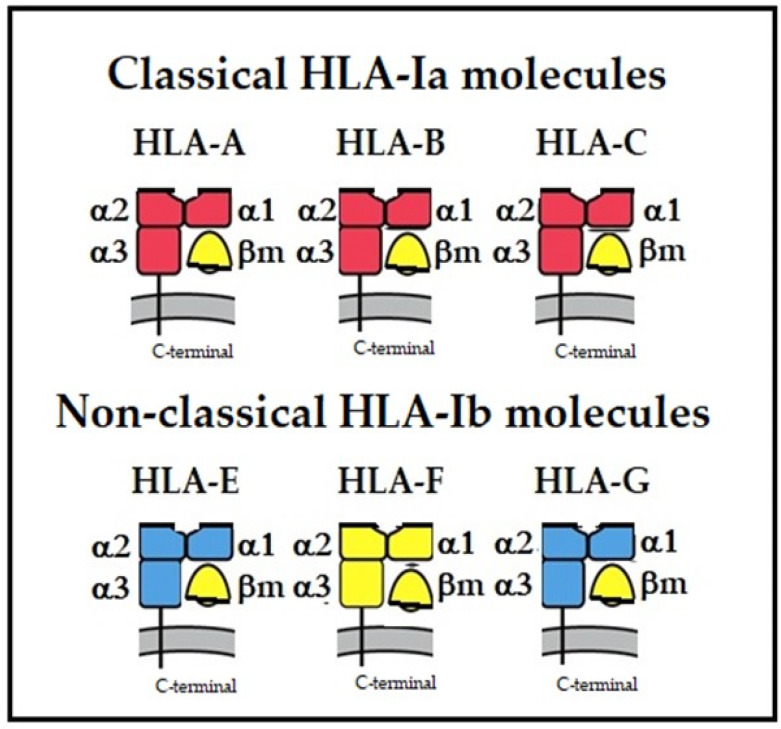
Isotypes of human major histocompatibility proteins. The glycosylated heavy chain (HC, 44 kDa) consists of the membrane distal G-ALPHA1 and G-ALPHA2 domains and of the proximal C-like ALPHA3 domain and is non-covalently associated with non-glycosylated B2m (11.6 kDa).

**Figure 2 biomolecules-13-01178-f002:**
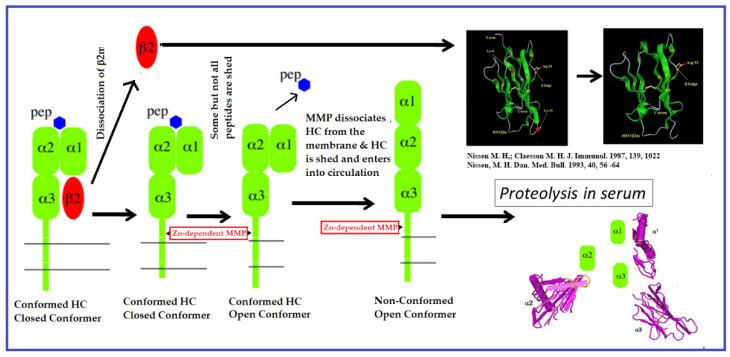
Events occurring after dissociation of B2m, based on the previous reports (Demaria et al. [11,12,13], Elliott et al. [14,15], Bix and Raulet, [17]). It has been well documented (Cerundolo et al. [16]) that the human mutant cell T2 expressing H-2D^b^ HCs, carrying 9 amino acid peptides, and complexed with human B2m, forms stable complexes with half-life > 110 h at 4 °C, 39 h at 22 °C, and 3 h at 37 °C. They have also shown that small increases in the length of the peptide greatly reduce the half life (t_½_ to about 1–10 h at 4 °C). The transformation of conformed HC to non-conformed HC and the cleavage of both conformed and non-conformed HC by MMPs and proteolytic degradation of B2m and HC (Demaria et al. [11,12]; Elliot, [14]). Proteolytic modifications of B2m in sera are based on Nissen et al. [18,19].

**Figure 3 biomolecules-13-01178-f003:**
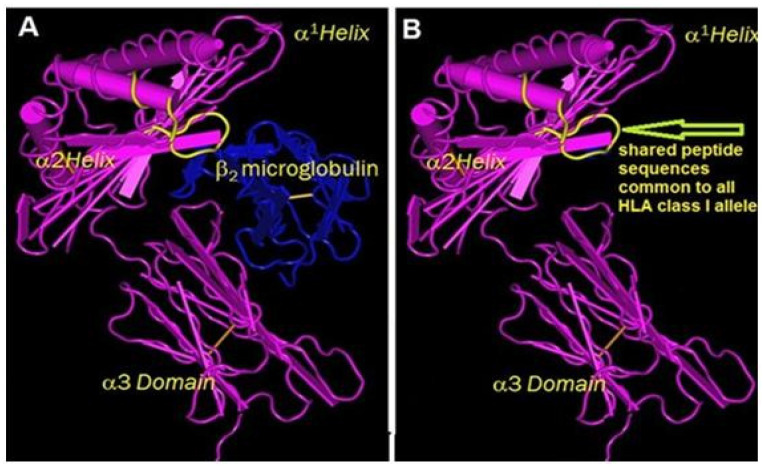
(**A**,**B**) Structure of HLA-I intact molecule (Face-1) showing sequences shared by all HLA loci (yellow string in α2 helical domain), which is masked by B2M (shown in blue). (**C**,**D**). The position of most commonly shared sequences are as follows: ^137^DTAAQI^142^ (present in HLA-B and HLA-C in G-ALPHA2 between the D strand and the helix, IMGT numbering 49–54, Lefranc et al. [3]) and ^115^QFAYDGKDY^123^ (present in all loci of HLA-Ia and Ib in G-ALPHA2 BC loop, IMGT numbering 25–33). The binding of mAb MEM-E/02 to HLA heavy chain coated solid matrix is inhibited by the above-mentioned peptides, Ravindranath et al. [37], shown as yellow curved line. In Face-1, these peptides are masked by B2m, but exposed in B2m-free HCs. The asterisk in figure show the sequence blocked by B2m in the native state.

**Figure 4 biomolecules-13-01178-f004:**
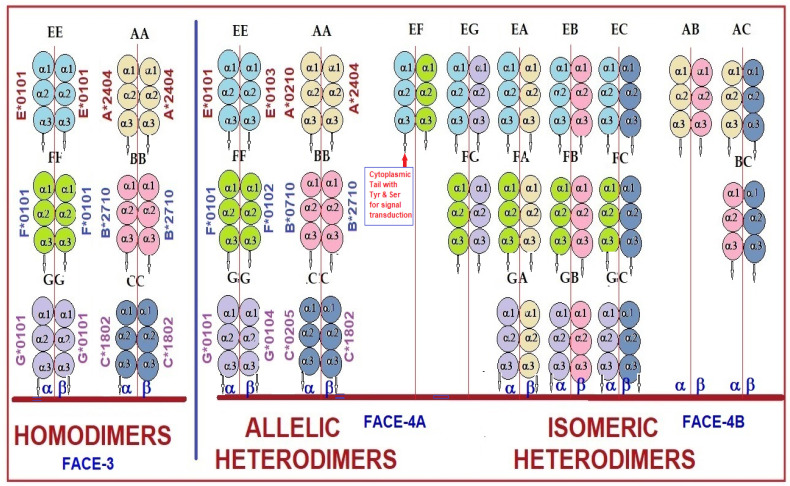
Diagrammatic illustration of different combinations of dimers of peptide-free (implying unfolded) monomeric B2m-free HCs (Face-2) resulting in different kinds of peptide-free (implying unfolded) allelic homodimers (Face-3), peptide-free (implying unfolded) allelic heterodimers (Face-4A), and peptide-free (implying unfolded) isomeric heterodimers (Face-4B). For the purpose of illustrating a similarity with HLA-II, we have suggestively and tentatively indicated one monomer as α chain and another monomer as β chain, although in reality both of them are α chains. As defined and designated by IMGT labels (Lefranc et al. [3]), each HC, though considered unfolded, may consist of groove (G)-like domains (α1 [D1] and α2 [D2]) and a constant (C)-like domain (α3 [D3]) with cytoplasmic tail. Domains of different alleles of different isomers or the α and β chains as shown in the figure may vary extensively. This implies that the ligands binding to the α1 domain of these two chains may differ markedly. More structural studies on the dimers may further clarify the G and C domains of each chain in a homodimer and in the allelic and isomeric heterodimers.

**Figure 5 biomolecules-13-01178-f005:**
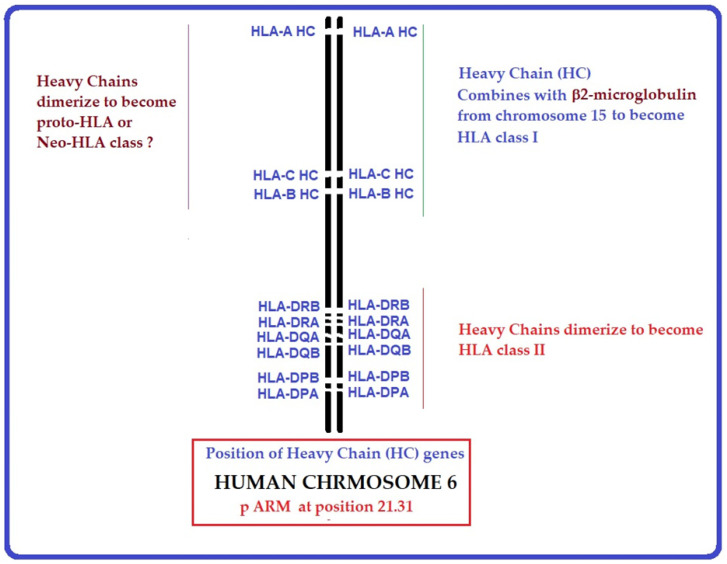
Illustration of the genes responsible for the heavy chains (HCs) of HLA classes. In HLA class I, the HCs generated by genes in chromosome 6 combine with B2m generated from the gene at chromosome 15, whereas HLA class II is a result of dimerization of HCs formed from genes in chromosome 6. The illustration guides to hypothesize the generation of a new class of HLAs (a neo-HLA class) from genes in chromosome 6 by their homo- or hetero-dimerization.

**Figure 6 biomolecules-13-01178-f006:**
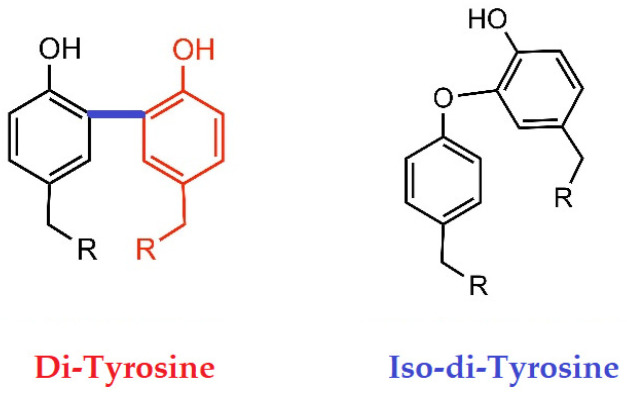
Formation of Di-tyrosine and Iso-di-tyrosine. Tyr phenoxyl radical can self-react to produce dityrosine within a sequence of HCs or between two HCs, depending on closer association of tyrosyl resides.

**Figure 7 biomolecules-13-01178-f007:**
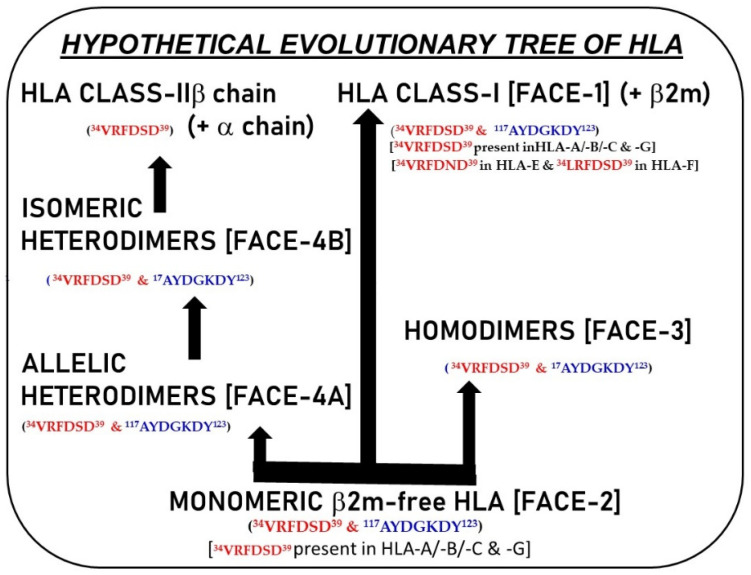
Hypothetical outline of the evolutionary tree of HLA. B2m-free HC of HLA-I or the proto-HLA is considered as the primitive variant of HLA-I and II, which by dimerization gives rise to homo- and heterodimers. B2m would have associated with monomers (Face-2) to evolve as HLA-I. Another independent monomer would have combined with modified Face-2 to evolve as HLA-II heterodimers.

**Table 1 biomolecules-13-01178-t001:** The HLA class I polyreactivity of commercial anti-HLA-E mAbs (MEM-E/02, MEM-E/06, MEM-E/07, MEM-E/08, 3D12), claimed to be HLA-E-specific, is compared with truly polyreactive mAb TFL-006 [47,48,49] in contrast to the HLA-I non-reactive, HLA-E monospecific mAb TFL-033 [40,41,42,43,44]. The HLA reactivity of the mAbs was examined with the single antigen bead assay using Luminex xMAP multiplex technology with dual-laser flow cytometry. The data are based on the LABScreen xMAP microbeads coated with an admixture of B2m-free and B2m-associated array of HLA antigens representing various alleles [45,46]. TFL-006 was also tested with LIFECODES xMAP microbeads coated only with intact B2m-associated HLA alleles [49] and all antigen-coated beads gave negative results, indicating that TFL-006 does not bind to Face-1.

HLA-Ia Alleles	mAb	mAb	mAb	mAb MEM Series				HLA-Ia Alleles	mAb	mAb	mAb	mAb MEM Series			
	TFL-033	TFL-006	3D12	E/02	E/06	E/07	E/08		TFL-033	TFL-006	3D12	E/02	E/06	E/07	E/08
	HLA-E specific	HLA-E non-specific mAbs							HLA-E specific	HLA-E non-specific mAbs					
**A*01:01**		1684			11,453			**B*07:02**		1080		1910	12,951	1375	618
**A*03:01**		649	789		9525			**B*08:01**		2422		1062	15,204		4281
**A*11:01**		4600		2940	7067	4992	1321	**B*13:01**		1956	1820	5700	15,934	6182	3195
**A*11:02**		607		559	9404	580		**B*13:02**		5604	1171	1326	16,249	393	
**A*23:01**					12,666			**B*14:01**		2919		3135	14,876	4103	1308
**A*24:02**		1256		4096	14,316	10,991	2739	**B*14:02**				942	12,447		634
**A*24:03**		1924		2505	15,894	9981	2094	**B*15:01**		3440		832	14,606		
**A*25:01**		788		629				**B*15:02**		1345		3250	15,676	3860	1051
**A*29:02**		1204		1593		1770	643	**B*15:03**		1874		4731	15,401		571
**A*30:01**		2232		526	10,072		551	**B*15:10**		1292		768	15,482		
**A*30:02**		1368			5296			**B*15:12**		2619		1903	15,670	850	
**A*32:01**		955		603				**B*15:13**		2833	591	3400	15,450	4023	982
**A*33:01**		1079		3037	1195	3469	1163	**B*15:16**		3503			13,159		
**A*33:03**		1204		1604		1228	511	**B*18:01**		1352		4392	16,138	3923	1665
**A*34:01**		2362		991	757	754		**B*27:05**		3518		942	13,985		
**A*36:01**		4993		1219	9451	1493		**B*27:08**		4668	3264	1175	14,289	640	
**A*66:01**		1783		571				**B*35:01**		2834	566	8716	15,768	12,917	6233
**A*68:01**		4150		664				**B*37:01**		1137		3444	14,356	3109	1871
**A*69:01**		992		917				**B*38:01**		6219	1672	968	14,774	516	
**C*01:02**		3238	966	3125	12,748	2998	3217	**B*39:01**		3301		3010	12,219	3825	1289
**C*02:02**		5570	720	2567	17,083	2003	1986	**B*40:01**		8083	800	3478	15,269	2662	
**C*03:02**		15,443		1713	14,941	1705	624	**B*40:02**		763	712	2442	9971	2166	631
**C*0303**		5990	571	2358	14,248	1549	1067	**B*40:06**		4678	3216	9898	15,642	14,269	6208
**C*03:04**		10,870		2585	13,686	1875	903	**B*41:01**		3037		4987	14,635	5400	1331
**C*04:03**		5809	3796	1765	7077	2052	703	**B*42:01**		2163			14,286		
**C*05:01**		3635	931	9263	15,923	15,435	6346	**B*44:02**		1537		2621	15,821	1525	
**C*06:02**		8171		3076	17,836	1914	10,260	**B*44:03**		1584	1321	2654	13,339	1625	
**C*07:02**		3082	1640	6680	1911	7149	13,655	**B*45:01**		4122	604	3134	14,671	3234	748
**C*08:01**		8093	592	2481	15,461	2001		**B*46:01**		4176		3042	16,173	3389	1062
**C*12:03**		9256	1020	1692	15,372	1035	771	**B*47:01**		2630		777	8849		
**C*14:02**		5143		1889	12,570	1839	1637	**B*48:01**		2822		3577	10,982		514
**C*15:02**		5101		2688	16,008	918	1080	**B*49:01**		3201		1588	15,804		
**C*16:01**		3833	530	1128	15,803	590	735	**B*50:01**		4895		769	15,575		
**C*17:01**		7054	5554	1869		1361	841	**B*51:01**		6464	841	2485	13,576	2619	884
**C*18:02**		1402	1095	7779	14,530	11,223	8373	**B*51:02**		3766		2303	13,138	2352	800
								**B*52:01**		1168	1416	928	8975		
								**B*53:01**		2324		2754	13,622	2890	1384
								**B*54:01**		1186		1910	13,506	1274	598
								**B*55:01**		3525		1287	14,420	892	180
								**B*56:01**		6569		5352	16,067	5075	1881
								**B*57:01**		4535	588	3626	11,746	5398	1982
								**B*57:03**		388	1143	2586	14,723	3272	1556
								**B*58:01**		2668	823	1636	11,721	1809	1155
								**B*59:01**		4289		2803	16,222	1837	915
								**B*67:01**		2308	1856	704	12,533		
								**B*73:01**		5661	659	5560	2363	8347	3629
								**B*78:01**		6082		4273	11,454	5927	1678
								**B*81:01**		7621	579	1097	11,758		
								**B*82:01**		4317		5295	15,480	6315	1922

**Table 2 biomolecules-13-01178-t002:** The citations referred to below on different cancers have used commercial mAbs MEM-E/02, MEM-E/06, MEM-E/07, and MEM-E/08 with the sole intention of identifying HLA-E. Most of these reports did not validate the reliability of the MEM series for HLA-E specificity.

Cancer Types	MEM Series	Citations
Melanoma	MEM-E/02	Derré L et al. [50]
Melanoma		Ravindranath M. H. et al. [36]
Melanoma		Allard M et al. [51]
Lip Squamosal cell carcinoma	MEM-E/06 & MEM-E/07	Goncalves A. S. et al. [52]
Laryngeal carcinoma		Silva T. G. et al. [53]
Vulvar intraepithelial carcinoma		van Esch E. M. G. et al. [54]
Penile carcinoma		Djajadiningrat R. S. et al. [55]
Glioblastoma	MEM-E/02	Mittelbronn, M. et al. [56]
Glioblastoma		Kren L et al. [57]
Glioblastoma		Kren L et al. [58]
Oral Osteosarcoma		Arantes D. A. C. et al. [59]
Intraoral mucoepidermoid carcinoma		Moscon C et al. [60]
Rectal Cancer		Reimers M. S. et al. [61]
Colorectal carcinoma	MEM-E/02	Benevolo M. et al. [62]
Colorectal carcinoma		Zeestraten E.C. et al. [63]
Colorectal carcinoma		Guo Z. Y. et al. [64]
Colorectal carcinoma		Huang R. et al. [65]
Colorectal carcinoma	MEM-E/08	Levy E. M. et al. [66]
Colorectal carcinoma		Levy, E. M. et al. [67]
Colon carcinoma and leukemia	MEM-E/02	Stangl S. et al. [68]
Gastric Cancer		Sasaki T. et al., [40]
Gastric Cancer		Ishigami S. et al. [69]
Hepatocellular carcinoma		Chen A. et al. [70]
Non-small cell Lung Carcinoma		Yazdi T. M. et al. [71]
Breast cancer		de Kruijf E. M. et al. [72]
Breast cancer		da Silva G. B. et al. [73]
Ovarian cancer/Cervical cancer		Gooden M et al. [74]
Cervical cancer		Gonçalves M. A. et al. [75]
Cervical cancer		Spaans V. M. et al., [76]
Cervical cancer		Ferns D. M. et al. [77]
Serous Ovarian Adenocarcinoma		Zheng H. et al. [78]
Serous Ovarian Adenocarcinoma		Andersson E. et al. [79]
Renal Cell Carcinoma		Hanak L. et al. [80]
Serous Ovarian Adenocarcinoma		Kren L. et al. [81]
Thyroid cancer		Zanetti B. R. et al. [82]
Hodgkin Lymphoma		Kren L, et al. [83]

**Table 3 biomolecules-13-01178-t003:** The distribution of Cys and Tyr in HLA-B alleles, based on the examination of a.a. sequences of α1, α2, and α4 domains of 847 B alleles. Cys is observed in 100% of alleles at position 101 and 164, 90% of alleles at position 67, and 50% of alleles in position 203. Compared to Cys, Tyr is almost 100% prevalent in positions 7, 9, 27, 59, 74, 84, 85, 99, 118, 123, 159, 171, and <50% in position 111, 116, and 209. Interestingly, Tyr is observed in four B*07 and in one B93 allele.

Allelles	α 1											α 2							α 3	
	1	7	9	27	59	67	74	84	85	99	101	113	116	118	123	159	164	171	203	209
**B*07:02**		Y	Y	Y	Y	Y		Y	Y	Y	C			Y	Y	Y	C	Y	C	Y
**B*07:09**		Y	Y	Y	Y	Y		Y	Y	Y	C			Y	Y	Y	C	Y		
**B *0710**		Y	Y	Y	Y	C		Y	Y	Y	C			Y	Y	Y	C	Y	C	Y
**B*07:14**		Y	Y	Y	Y	Y		Y	Y	Y	C	Y	Y	Y	Y	Y	C	Y		
**B*07:17**		Y	Y	Y	Y	Y		Y	Y	Y	C	Y		Y	Y	Y	C	Y	C	Y
**B*14:01**		Y	Y	Y	Y	C		Y	Y	Y	C	Y		Y	Y	Y	C		C	Y
**B*14:06**		Y	Y	Y	Y	C		Y	Y	Y	C	Y		Y	Y	Y	C			
**B*15:02**		Y	Y	Y	Y		Y	Y	Y	Y	C	Y		Y	Y	Y	C	Y	C	Y
**B*15:09**		Y	Y	Y	Y	C	Y	Y	Y	Y	C			Y	Y	Y	C	Y	C	Y
**B*15:10**		Y	Y	Y	Y	C	Y	Y	Y	Y	C		Y	Y	Y	Y	C	Y	C	Y
**B*15:18**		Y	Y	Y	Y	C	Y	Y	Y	Y	C		Y	Y	Y	Y	C	Y	C	Y
**B*15:21**		Y	Y	Y	Y	C	Y	Y	Y	Y	C	Y		Y	Y	Y	C	Y	C	Y
**B*15:23**		Y	Y	Y	Y	C	Y	Y	Y	Y	C			Y	Y	Y	C	Y	C	Y
**B*15:44**		Y	Y	Y	Y	C	Y	Y	Y	Y	C	Y		Y	Y	Y	C	Y		
**B*15:45**		Y	Y	Y	Y		Y	Y	Y	Y	C	Y	Y	Y	Y	Y	C	Y		
**B*15:90**		Y	Y	Y	Y	C	Y	Y	Y	Y	C		Y	Y	Y	Y	C	Y		
**B*15:91**		Y	Y	Y	Y		Y	Y	Y	Y	C			Y	Y	Y	C	Y		
**B*15:92**		Y	Y	Y	Y		Y	Y	Y	Y	C			Y	Y	Y	C	Y	C	Y
**B*15:93**		Y	Y	Y	Y	C	Y	Y	Y	Y	C			Y	Y	Y	C	Y	C	Y
**B*15:99**		Y	Y	Y	Y	C	Y	Y	Y	Y	C		Y	Y	Y	Y	C	Y		
**B*27:01**		Y		Y	Y	C		Y	Y	Y	C	Y		Y	Y	Y	C	Y		
**B*27:02**		Y		Y	Y	C		Y	Y	Y	C	Y		Y	Y	Y	C	Y	C	Y
**B*27:03**		Y		Y	Y	C		Y	Y	Y	C	Y		Y	Y	Y	C	Y	C	Y
**B*27:04:01**		Y		Y	Y	C		Y	Y	Y	C	Y		Y	Y	Y	C	Y	C	Y
**B*27:04:02**		Y		Y	Y	C		Y	Y	Y	C	Y		Y	Y	Y	C	Y	C	
**B*27:05:02**		Y		Y	Y	C		Y	Y	Y	C	Y		Y	Y	Y	C	Y	C	Y
**B*27:05:03**		Y		Y	Y	C		Y	Y	Y	C	Y		Y	Y	Y	C	Y		
**B*27:05:04**		Y		Y	Y	C		Y	Y	Y	C	Y		Y	Y	Y	C	Y		
**B*27:05:05**		Y		Y	Y	C		Y	Y	Y	C	Y		Y	Y	Y	C	Y	C	Y
**B*27:05:06**		Y		Y	Y	C		Y	Y	Y	C	Y		Y	Y	Y	C	Y		
**B*27:05:07**		Y		Y	Y	C		Y	Y	Y	C	Y		Y	Y	Y	C	Y		
**B*27:05:08**		Y		Y	Y	C		Y	Y	Y	C	Y		Y	Y	Y	C	Y		
**B*27:05:09**		Y		Y	Y	C		Y	Y	Y	C	Y		Y	Y	Y	C	Y		
**B*27:05**		Y		Y	Y	C		Y	Y	Y	C	Y		Y	Y	Y	C	Y		
**B*27:06**		Y		Y	Y	C		Y	Y	Y	C	Y	Y	Y	Y	Y	C	Y	C	
**B*27:07**		Y		Y	Y	C		Y	Y	Y	C		Y	Y	Y	Y	C	Y	C	
**B*27:08**		Y		Y	Y	C		Y	Y	Y	C	Y		Y	Y	Y	C	Y	C	
**B*27:09**		Y		Y	Y	C		Y	Y	Y	C	Y		Y	Y	Y	C	Y	C	
**B*27:10**		Y		Y	Y	C		Y	Y	Y	C	Y		Y	Y	Y	C	Y		
**B*27:11**		Y		Y	Y	C		Y	Y	Y	C		Y	Y	Y	Y	C	Y	C	
**B*27:12**		Y		Y	Y	C		Y	Y	Y	C	Y		Y	Y	Y	C	Y	C	
**B*27:13**		Y		Y	Y	C		Y	Y	Y	C	Y		Y	Y	Y	C	Y	C	
**B*27:14**		Y		Y	Y	C		Y	Y	Y	C	Y		Y	Y	Y	C	Y		
**B*27:15**		Y		Y	Y	C		Y	Y	Y	C	Y		Y	Y	Y	C	Y		
**B*27:16**		Y		Y	Y	C		Y	Y	Y	C	Y		Y	Y	Y	C	Y		
**B*27:17**		Y		Y	Y	C		Y	Y	Y	C	Y		Y	Y	Y	C	Y		Y
**B*27:18**		Y		Y	Y		Y	Y	Y	Y	C	Y		Y	Y	Y	C	Y		
**B*27:19**		Y		Y	Y	C		Y	Y	Y	C	Y		Y	Y	Y	C	Y		
**B*27:20**		Y		Y	Y	C		Y	Y	Y	C		Y	Y	Y	Y	C	Y		Y
**B*27:21**		Y		Y	Y	C		Y	Y	Y	C	Y	Y	Y	Y	Y	C	Y		Y
**B*27:23**		Y		Y	Y		Y	Y	Y	Y	C			Y	Y	Y	C	Y		
**B*27:24**		Y		Y	Y	C		Y	Y	Y	C	Y	Y	Y	Y	Y	C	Y		
**B*27:25**		Y		Y	Y	C		Y	Y	Y	C	Y		Y	Y	Y	C	Y		
**B*27:26**		Y		Y	Y	C		Y	Y	Y	C	Y		Y	Y	Y	C	Y		
**B*27:27**		Y		Y	Y	C		Y	Y	Y	C		Y	Y	Y	Y	C	Y		
**B*27:28**		Y		Y	Y	C		Y	Y	Y	C	Y		Y	Y	Y	C			
**B*27:29**		Y		Y	Y		Y	Y	Y	Y	C	Y		Y	Y	Y	C	Y		
**B*27:30**		Y		Y	Y	C		Y	Y	Y	C	Y		Y	Y	Y	C	Y		
**B*27:31**		Y		Y	Y	C		Y	Y	Y	C	Y		Y	Y	Y	C	Y		
**B*27:32**		Y		Y	Y	C		Y	Y	Y	C	Y		Y	Y	Y	C	Y	C	
**B*27:33**		Y		Y	Y	C		Y	Y	Y	C		Y	Y	Y	Y	C	Y		
**B*27:34**		Y		Y	Y	C		Y	Y	Y	C		Y	Y	Y	Y	C	Y		
**B*27:35**		Y		Y	Y	C		Y	Y	Y	C	Y		Y	Y	Y	C	Y	C	
**B*27:36**		Y		Y	Y	C		Y	Y	Y	C	Y		Y	Y	Y	C	Y	C	
**B*35:26**		Y	Y	Y	Y	C	Y	Y	Y	Y	C			Y	Y	Y	C	Y		
**B*38:01:01**		Y	Y	Y	Y	C	Y	Y	Y	Y	C			Y	Y	Y	C	Y	C	Y
**B*38:01:02**		Y	Y	Y	Y	C	Y	Y	Y	Y	C			Y	Y	Y	C	Y		
**B*38:02:01**		Y	Y	Y	Y	C	Y	Y	Y	Y	C			Y	Y	Y	C	Y		
**B*38:02:02**		Y	Y	Y	Y	C	Y	Y	Y	Y	C			Y	Y	Y	C	Y		
**B*38:03**		Y	Y	Y	Y			Y	Y	Y	C			Y	Y	Y	C	Y		
**B*38:04**		Y	Y	Y	Y	C	Y	Y	Y	Y	C			Y	Y	Y	C	Y		
**B*38:05**		Y	Y	Y	Y	C	Y	Y	Y	Y	C			Y	Y	Y	C	Y	C	Y
**B*38:06**		Y	Y	Y	Y	C	Y	Y	Y	Y	C			Y	Y	Y	C	Y		
**B*38:07**		Y	Y	Y	Y	C	Y	Y	Y	Y	C			Y	Y	Y	C	Y		
**B*38:08**		Y	Y	Y	Y	C	Y	Y	Y	Y	C			Y	Y	Y	C	Y		
**B*38:09**		Y	Y	Y	Y	C		Y	Y		C			Y	Y	Y	C	Y		
**B*38:10**		Y	Y	Y	Y	C	Y	Y	Y	Y	C			Y	Y	Y	C	Y		
**B*38:11**		Y	Y	Y	Y	C	Y	Y	Y	Y	C			Y	Y	Y	C	Y		
**B*38:12**		Y	Y	Y	Y	C	Y	Y	Y	Y	C			Y	Y	Y	C	Y	C	Y
**B*38:13**		Y	Y	Y	Y	C	Y	Y	Y	Y	C			Y	Y	Y	C	Y		
**B*38:14**		Y	Y	Y	Y	C		Y	Y	Y	C			Y	Y	Y	C	Y	C	Y
**B*38:15**		Y	Y	Y	Y	C	Y	Y	Y	Y	C			Y	Y	Y	C	Y		
**B*39:01:01:01**		Y	Y	Y	Y			Y	Y	Y	C			Y	Y	Y	C	Y	C	Y
**B*39:01:01:02**		Y	Y	Y	Y			Y	Y	Y	C			Y	Y	Y	C	Y	C	Y
**B*39:01:03**		Y	Y	Y	Y	C		Y	Y	Y	C			Y	Y	Y	C	Y	C	Y
**B*39:01:04**		Y	Y	Y	Y	C		Y	Y	Y	C			Y	Y	Y	C	Y		
**B*39:02:01**		Y	Y	Y	Y	C		Y	Y	Y	C			Y	Y	Y	C	Y	C	Y
**B*39:02:02**		Y	Y	Y	Y	C		Y	Y	Y	C			Y	Y	Y	C	Y	C	Y
**B*39:03**		Y	Y	Y	Y	C		Y	Y	Y	C			Y	Y	Y	C	Y	C	Y
**B*39:04**		Y	Y	Y	Y	C		Y	Y	Y	C			Y	Y	Y	C	Y	C	Y
**B*39:05**		Y	Y	Y	Y	C	Y	Y	Y	Y	C			Y	Y	Y	C	Y	C	Y
**B*39:06:01**		Y	Y	Y	Y	C		Y	Y	Y	C			Y	Y	Y	C	Y	C	Y
**B*39:06:02**		Y	Y	Y	Y	C		Y	Y	Y	C			Y	Y	Y	C	Y	C	Y
**B*39:07**		Y	Y	Y	Y	C	Y	Y	Y	Y	C			Y	Y	Y	C	Y		
**B*39:08**		Y	Y	Y	Y		Y	Y	Y	Y	C			Y	Y	Y	C	Y	C	Y
**B*39:09**		Y	Y	Y	Y	C		Y	Y	Y	C			Y	Y	Y	C	Y	C	Y
**B*39:10**		Y	Y	Y	Y	C		Y	Y	Y	C			Y	Y	Y	C	Y	C	Y
**B*39:11**		Y	Y	Y	Y	C	Y	Y	Y	Y	C			Y	Y	Y	C	Y	C	Y
**B*39:12**		Y	Y	Y	Y	C		Y	Y	Y	C			Y	Y	Y	C	Y		
**B*39:13:L01**		Y	Y	Y	Y		Y	Y	Y	Y	C			Y	Y	Y	C	Y		
**B*39:13:02**		Y	Y	Y	Y		Y	Y	Y	Y	C			Y	Y	Y	C	Y	C	Y
**B*39:14**		Y	Y	Y	Y	C		Y	Y	Y	C			Y	Y	Y	C	Y		
**B*39:15**		Y	Y	Y	Y	C		Y	Y	Y	C			Y	Y	Y	C	Y		
**B*39:16**		Y	Y	Y	Y	C		Y	Y	Y	C			Y	Y	Y	C	Y		
**B*39:17**		Y	Y	Y	Y	C		Y	Y	Y	C	Y		Y	Y	Y	C	Y		
**B*39:18**		Y	Y	Y	Y	C		Y	Y	Y	C			Y	Y	Y	C	Y		
**B*39:19**		Y	Y	Y	Y	C		Y	Y	Y	C			Y	Y	Y	C	Y		
**B*39:20**		Y	Y	Y	Y	C	Y	Y	Y	Y	C			Y	Y	Y	C	Y		
**B*39:22**		Y	Y	Y	Y	C		Y	Y	Y	C			Y	Y	Y	C	Y		
**B*39:23**		Y	Y	Y	Y			Y	Y	Y	C			Y	Y	Y	C	Y	C	Y
**B*39:24**		Y	Y	Y	Y	C		Y	Y	Y	C			Y	Y	Y	C	Y	C	Y
**B*39:25N**		Y	Y	Y	Y	C		Y	Y	Y	C									
**B*39:26**		Y	Y	Y	Y	C		Y	Y	Y	C			Y	Y	Y	C	Y		
**B*39:27**		Y	Y	Y	Y	C		Y	Y	Y	C			Y	Y	Y	C	Y		
**B*39:28**		Y	Y	Y	Y	C		Y	Y	Y	C			Y	Y	Y	C	Y		
**B*39:29**		Y	Y	Y	Y	C		Y	Y	Y	C		Y	Y	Y	Y	C	Y		
**B*39:30**		Y	Y	Y	Y	C		Y	Y	Y	C			Y	Y	Y	C	Y		
**B*39:31**		Y	Y	Y	Y	C		Y	Y	Y	C			Y	Y	Y	C	Y		
**B*39:32**		Y	Y	Y	Y	C		Y	Y	Y	C			Y	Y	Y	C			
**B*39:33**		Y	Y	Y	Y	C		Y	Y	Y	C			Y	Y	Y	C	Y	C	Y
**B*39:34**		Y	Y	Y	Y	C		Y	Y	Y	C			Y	Y	Y	C	Y	C	Y
**B*39:35**		Y	Y	Y	Y	C		Y	Y	Y	C			Y	Y	Y	C	Y		
**B*39:36**		Y	Y	Y	Y	C		Y	Y	Y	C			Y	Y	Y	C	Y	C	Y
**B*39:37**		Y	Y	Y	Y	C	Y	Y	Y	Y	C			Y	Y	Y	C	Y		
**B*39:38Q**		Y	Y	Y	Y	C		Y	Y	Y	C			Y	Y	Y		Y	C	Y
**B*39:39**		Y	Y	Y	Y			Y	Y	Y	C			Y	Y	Y	C	Y		
**B*39:40N**		Y	Y	Y	Y	C		Y	Y	Y	C			Y	Y	Y				
**B*39:41**		Y	Y	Y	Y	C		Y	Y	Y	C			Y	Y	Y	C	Y	C	Y
**B*73:01**		Y	Y	Y	Y	C		Y	Y	Y	C	Y		Y	Y	Y	C		C	Y
**B*78:03**		Y	Y	Y	Y	C		Y	Y	Y	C		Y	Y	Y	Y	C			
**B*95:01**		Y	Y	Y	Y	Y		Y	Y	Y	C			Y	Y	Y	C	Y		
**B*95:08**		Y	Y	Y	Y	C	Y	Y	Y	Y	C			Y	Y	Y	C	Y	C	Y
**B*95:09**		Y	Y	Y	Y		Y	Y	Y	Y	C			Y	Y	Y	C	Y		
**B*95:12**		Y	Y	Y	Y		Y	Y	Y	Y	C	Y		Y	Y	Y	C	Y		
**B*95:13**		Y	Y	Y	Y		Y	Y	Y	Y	C			Y	Y	Y	C	Y		
**B*95:14**		Y	Y	Y	Y	C	Y	Y	Y	Y	C			Y	Y	Y	C	Y	C	Y
**B*95:15**		Y	Y	Y	Y	C	Y	Y	Y	Y	C			Y	Y	Y	C	Y		
**B*95:19**		Y	Y	Y	Y	C	Y	Y	Y	Y	C			Y	Y	Y	C	Y		

**Table 4 biomolecules-13-01178-t004:** Distribution of Cys and Tyr in a few HLA-A alleles, based on the examination of a.a. sequences of α1, α2, and α4 domains of 505 A alleles. Cys is observed in 4 alleles at position 99, in almost all alleles at position 101 and 164, and in >50% of alleles at position 203. Compared with Cys, Tyr is almost 75 to 100% prevalent in positions 7, 9, 27, 59, 74, 84, 85, 99, 113, 116, 118, 123, 159, 171, and 209, and <50% in position 99.

ALLELES	α1										α2							α3	
	1	7	9	27	59	67	84	85	99	101	113	116	118	123	159	164	171	203	209
**E*01:01:0101**		Y		Y	Y		Y	Y		C	Y		Y	Y	Y	C	Y	C	Y
**A*01:01**		Y		Y	Y		Y	Y	Y	C	Y		Y	Y	Y	C	Y	C	Y
**A*01:04**		Y		Y	Y		Y	Y	Y	C	Y		Y	Y	Y	C	Y		
**A*02:01**		Y		Y	Y		Y	Y	C	C	Y	Y	Y	Y	Y	C	Y	C	Y
**A*02:07**		Y		Y	Y		Y	Y	C	C	Y	Y	Y	Y	Y	C	Y	C	Y
**A*02:15**		Y		Y	Y		Y	Y	C	C	Y	Y	Y	Y	Y	C	Y	C	Y
**A*02:18**		Y		Y	Y		Y	Y	C	C	Y	Y	Y	Y	Y	C	Y	C	Y
**A*02:10**		Y	Y	Y	Y		Y	Y		C	Y	Y	Y	Y	Y	C	Y	C	Y
**A*23:01**		Y		Y	Y		Y	Y		C	Y	Y	Y	Y	Y	C	Y	C	Y
**A*23:02**		Y		Y	Y		Y	Y		C	Y	Y	Y	Y	Y	C	Y		
**A*24:62**		Y		Y	Y		Y	Y		C	Y		Y	Y	Y	C	Y		
**A*68:02:01:01**		Y	Y	Y	Y		Y	Y	Y	C	Y	Y	Y	Y	Y	C	Y	C	Y
**A*68:15**		Y	Y	Y	Y		Y	Y	Y	C	Y	Y	Y	Y	Y	C	Y		
**A*92:03**		Y		Y	Y		Y	Y	C	C	Y	Y	Y	Y	Y	C	Y	C	Y

**Table 5 biomolecules-13-01178-t005:** The distribution of Cys and Tyr in HLA-C alleles, based on the examination of a. a sequences of α1, α2, and α4 domains of all C alleles. Cys is lacking at position 67 and at 99 (with the exception of C*01:02:01, C*01:03, and C*01:04), and is present in almost all alleles at position 101, 164, and 203. Compared to Cys, Tyr is almost 75 to 100% prevalent in positions 7, 9, 27, 59, 74, 84, 85, 99, 113, 118, 123, 159, 171, and 209.

ALLELES	α1										α 2							α 3	
	1	7	9	27	58	67	84	85	99	101	113	116	118	123	159	164	171	203	209
**E*01:01:01:01**	C	Y		Y			Y	Y		C	Y		Y	Y	Y	C	Y	C	Y
**C*01:02:01**	C	Y		Y	Y	Y	Y	Y	C	C	Y	Y	Y	Y	Y	C	Y	C	Y
**C*01:03**	C	Y		Y	Y	Y	Y	Y	C	C	Y		Y	Y	Y	C	Y	C	Y
**C*01:04**	C	Y		Y	Y	Y	Y	Y	C	C	Y		Y	Y	Y	C	Y	C	Y
**C*02:02:01**	C	Y	Y	Y	Y	Y	Y	Y	Y	C	Y		Y	Y	Y	C	Y	C	Y
**C*02:02:02**	C	Y	Y	Y	Y	Y	Y	Y	Y	C	Y		Y	Y	Y	C	Y	C	Y
**C*02:10**	C	Y	Y	Y	Y	Y	Y	Y	Y	C	Y		Y	Y	Y	C	Y	C	Y
**C*02:11**	C	Y	Y	Y	Y	Y	Y	Y	Y	C	Y		Y	Y	Y	C	Y	C	Y
**C*05:01:01**	C	Y	Y	Y	Y	Y	Y	Y	Y	C	Y		Y	Y	Y	C	Y	C	Y
**C*05:03**	C	Y	Y	Y	Y	Y	Y	Y	Y	C	Y		Y	Y	Y	C	Y	C	Y
**C*05:08**	C	Y	Y	Y	Y	Y	Y	Y	Y	C	Y		Y	Y	Y	C	Y	C	Y
**C*05:09**	C	Y	Y	Y	Y	Y	Y	Y	Y	C	Y		Y	Y	Y	C	Y	C	Y
**C*06:02:01:01**	C	Y		Y	Y	Y	Y	Y	Y	C	Y		Y	Y	Y	C	Y	C	Y
**C*06:02:01:02**	C	Y		Y	Y	Y	Y	Y	Y	C	Y		Y	Y	Y	C	Y	C	Y
**C*07:01:01**	C	Y		Y	Y	Y	Y	Y	Y	C	Y		Y	Y	Y	C	Y	C	Y
**C*07:01:02**	C	Y		Y	Y	Y	Y	Y	Y	C	Y		Y	Y	Y	C	Y	C	Y
**C*07:02:01:01**	C	Y		Y	Y	Y	Y	Y		C	Y		Y	Y	Y	C	Y	C	Y
**C*07:02:01:02**	C	Y		Y	Y	Y	Y	Y		C	Y		Y	Y	Y	C	Y	C	Y
**C*07:02:01:03**	C	Y		Y	Y	Y	Y	Y		C	Y		Y	Y	Y	C	Y	C	Y
**C*07:03**	C	Y		Y	Y	Y	Y	Y		C	Y		Y	Y	Y	C	Y	C	Y
**C*07:04:01**	C	Y		Y	Y	Y	Y	Y	Y	C	Y		Y	Y	Y	C	Y	C	Y
**C*07:04:02**	C	Y		Y	Y	Y	Y	Y	Y	C	Y		Y	Y	Y	C	Y	C	Y
**C*07:06**	C	Y		Y	Y	Y	Y	Y	Y	C	Y		Y	Y	Y	C	Y	C	Y
**C*07:11**	C	Y		Y	Y	Y	Y	Y	Y	C	Y		Y	Y	Y	C	Y	C	Y
**C*07:18**	C	Y		Y	Y	Y	Y	Y	Y	C	Y		Y	Y	Y	C	Y	C	Y
**C*07:19**	C	Y		Y	Y	Y	Y	Y		C	Y		Y	Y	Y	C	Y	C	Y
**C*07:26**	C	Y	Y	Y	Y	Y	Y	Y	Y	C	Y		Y	Y	Y	C	Y	C	Y
**C*07:29**	C	Y		Y	Y	Y	Y	Y		C	Y		Y	Y	Y	C	Y	C	Y
**C*07:30**	C	Y		Y	Y	Y	Y	Y	Y	C	Y		Y	Y	Y	C	Y	C	Y
**C*07:36**	C	Y		Y	Y	Y	Y	Y	Y	C	Y		Y	Y	Y	C	Y	C	Y
**C*07:41**	C	Y		Y	Y	Y	Y	Y	Y	C	Y		Y	Y	Y	C	Y	C	Y
**C*07:42**	C	Y		Y	Y	Y	Y	Y		C	F		Y	Y	Y	C	Y	C	Y
**C*08:01:01**	C	Y	Y	Y	Y	Y	Y	Y	Y	C	Y		Y	Y	Y	C	Y	C	Y
**C*08:02**	C	Y	Y	Y	Y	Y	Y	Y	Y	C	Y		Y	Y	Y	C	Y	C	Y
**C*08:03**	C	Y	Y	Y	Y	Y	Y	Y	Y	C	Y		Y	Y	Y	C	Y	C	Y
**C*08:09**	C	Y	Y	Y	Y	Y	Y	Y	Y	C	Y		Y	Y	Y	C	Y	C	
**C*12:02:01**	C	Y	Y	Y	Y	Y	Y	Y	Y	C	Y		Y	Y	Y	C	Y	C	Y
**C*12:02:02**	C	Y	Y	Y	Y	Y	Y	Y	Y	C	Y		Y	Y	Y	C	Y	C	Y
**C*12:03:01:01**	C	Y	Y	Y	Y	Y	Y	Y	Y	C	Y		Y	Y	Y	C	Y	C	Y
**C*12:04:01**	C	Y	Y	Y	Y	Y	Y	Y	Y	C	Y		Y	Y	Y	C	Y	C	
**C*12:04:02**	C	Y	Y	Y	Y	Y	Y	Y	Y	C	Y		Y	Y	Y	C	Y	C	Y
**C*12:05**	C	Y	Y	Y	Y	Y	Y	Y	Y	C	Y		Y	Y	Y	C	Y	C	Y
**C*12:08**	C	Y	Y	Y	Y	Y	Y	Y	Y	C	Y		Y	Y	Y	C	Y	C	Y
**C*12:13**	C	Y	Y	Y	Y	Y	Y	Y	Y	C	Y		Y	Y	Y	C	Y	C	Y
**C*12:14**	C	Y	Y	Y	Y	Y	Y	Y	Y	C	Y		Y	Y	Y	C	Y	C	Y
**C*12:19**	C	Y	Y	Y	Y	Y	Y	Y	Y	C	Y		Y	Y	Y	C	Y	C	Y
**C*14:02:01**	C	Y		Y	Y	Y	Y	Y		C	Y		Y	Y	Y	C	Y	C	Y
**C*14:03**	C	Y		Y	Y	Y	Y	Y		C	Y		Y	Y	Y	C	Y	C	Y
**C*15:02:01**	C	Y	Y	Y	Y	Y	Y	Y	Y	C			Y	Y	Y	C	Y	C	Y
**C*15:03**	C	Y	Y	Y	Y	Y	Y	Y	Y	C			Y	Y	Y	C	Y	C	Y
**C*15:04**	C	Y	Y	Y	Y	Y	Y	Y	Y	C	Y		Y	Y	Y	C	Y	C	Y
**C*15:05:01**	C	Y	Y	Y	Y	Y	Y	Y	Y	C			Y	Y	Y	C	Y	C	Y
**C*15:05:02**	C	Y	Y	Y	Y	Y	Y	Y	Y	C			Y	Y	Y	C	Y	C	Y
**C*15:06**	C	Y	Y	Y	Y	Y	Y	Y	Y	C			Y	Y	Y	C	Y	C	Y
**C*15:16**	C	Y	Y	Y	Y	Y	Y	Y	Y	C			Y	Y	Y	C	Y	C	Y
**C*15:17**	C	Y	Y	Y	Y	Y	Y	Y	Y	C			Y	Y	Y	C	Y	C	Y
**C*16:01:01**	C	Y	Y	Y	Y	Y	Y	Y	Y	C	Y		Y	Y	Y	C	Y	C	Y
**C*16:01:02**	C	Y	Y	Y	Y	Y	Y	Y	Y	C	Y		Y	Y	Y	C	Y		
**C*16:02**	C	Y	Y	Y	Y	Y	Y	Y	Y	C	Y		Y	Y	Y	C	Y	C	Y
**C*16:04:01**	C	Y	Y	Y	Y	Y	Y	Y	Y	C	Y		Y	Y	Y	C	Y	C	Y
**C*17:01**	G	Y	Y	Y	Y	Y	Y	Y	Y	C	Y		Y	Y	Y	C	Y	C	Y
**C*17:02**	G	Y	Y	Y	Y	Y	Y	Y	Y	C	Y		Y	Y	Y	C	Y	C	Y
**C*17:03**	G	Y	Y	Y	Y	Y	Y	Y	Y	C	Y		Y	Y	Y	C	Y	C	Y
**C*18:01**	C	Y		Y	Y	Y	Y	Y		C	Y		Y	Y	Y	C	Y	C	Y
**C*18:02**	C	Y		Y	Y	Y	Y	Y		C	Y		Y	Y	Y	C	Y	C	Y
**C*05:01:01**	C	Y	Y	Y	Y	Y	Y	Y	Y	C	Y		Y	Y	Y	C	Y	C	Y
**C*07:01:01**	C	Y		Y	Y	Y	Y	Y	Y	C	Y		Y	Y	Y	C	Y	C	Y
**C*18:01**	C	Y		Y	Y	Y	Y	Y		C	Y		Y	Y	Y	C	Y	C	Y

**Table 6 biomolecules-13-01178-t006:** Similarities in the a. a sequence of Proto-HLA, HLA class I heavy chains and β-chains of HLA-II.

**HLA class I and Proto-HLA** ** * ^34^ * ** ** *V R F D S D . . . . . . E . R A . . . . . . . . E Y W ^60^* **
^V: VALINE (NON-POLAR); R: ARGININE (+ CHARGE); F: PHENYLALANINE (NON-POLAR); D: ASPARTIC ACID (- CHARGE); S: SERINE (POLAR); E: GLUTAMIC ACID (- CHARGE); A: ALANINE (NON-POLAR); Y: TYROSINE (POLAR); W: TRYPTOPHAN (NON-POLAR)^
ALLELES EXAMINED for Heavy Chains *HLA-A* ALLELES 1-504 *HLA-B* ALLELES 1-846 *HLA-C* ALLELES 1-277 *HLA-G ALLELES 4* *HLA-E ALLELES 2 **^34^V R F D*** N ***D . . . . . .*** V ***. R A . . . . . . . . E Y W ^60^*** *HLA-F ALLELE 1 **^34^L R F D S D . . . . . . E . R*** E ***. . . . . . . .*** Q ***Y W ^60^***
**HLA class II** ***^38^**V R F D S D . . E . R A . . . . . . . . . E Y W ^61^*** ALLELES EXAMINED for β-chain of HLA-II *DRB* ALLELES 1-512 DRB1*010101–1*030101–1*040101–1*070101–1*080101–1*090102–*110101, DRB1*130101–1*140101–1*150101–1*160101, DRB3*0106, DRB5*0102, DRB5*0202–5*0205; *DQB* ALLELES 1–71 DQB1*050101–DQB1*020101, DQB1*060101 ***^36^**V R F D S D . . E . R A . . . . . . . . . E Y W ^59^*** *DPB* ALLELES 1–123 DPB1*020102–1*030101–1*0402–1*0501–1*0601–1*0801–1*0901–1*1001, DPB1*1102, DPB1*1401, DPB1*1502, DPB1*1601–1*1701–1*1801–1*1901–1*200101 ---- 1*9901

## Data Availability

All of the original data presented as tables and figures are available from the first author.

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
