# Peer review of "Cell Surface B2m-Free Human Leukocyte Antigen (HLA) Monomers and Dimers: Are They Neo-HLA Class and Proto-HLA?"

_biomolecules, 2023, doi:10.3390/biom13081178_

Round 1
Reviewer 1 Report
Ravindranath et al. describe in this review the current knowledge about b2m-free MHC-I molecules on the surface of cells. It is a really comprehensive and impressive opus and fascinating to see the historical aspects as well as the more recent developments in the field. The conclusion or speculation that free heavy chains of MHC-I may represent the "progenitor" of both HLA-types is supported by the current knowledge but also somehow provocative and future experiments will elucidate this idea and concept.
Author Response
Reply to Rev-1
Comment:
Ravindranath et al. describe in this review the current knowledge about b2m-free MHC-I molecules on the surface of cells. It is a really comprehensive and impressive opus and fascinating to see the historical aspects as well as the more recent developments in the field. The conclusion or speculation that free heavy chains of MHC-I may represent the "progenitor" of both HLA-types is supported by the current
Response:
We thank the reviewer for considering that the review is comprehensive and impressive opus and fascinating to see the historical aspects. Based on the Academic reviewer’s comments and the other two reviewers we have taken a significant effort to revise the manuscript extensively.
We are very much delighted to note that the reviewer concurs with our suggestion that B2m-free HCs may represent the “progenitor” of both HLA class I and II and thanks for noting that the current review supports it.
Reviewer 2 Report
The authors wrote a comprehensive review that is very well organized in structure and thought, especially the helpful "inferences" statements. The authors make compelling arguments that the concept(s) of HLA class I monomers and dimers that exist in pathological conditions such as cancer and autoimmunity are worthy of further investigations. As this review was quite thorough in both reviewing literature and the authors' genuine interpretations, the following minor comments are designed to add further value.
1. An issue with the manuscript was the excessive formatting errors throughout, especially Greek symbols that challenged the reviewer. In addition, 2 errors in word choice are "grove" instead of "groove", line 58 and "keep" instead of "kept", line 937.
2. Due et al demonstrated the "insulin receptor" associated with HLA-I HC molecules, not "insulin", lines 71-75.
3. Could the authors offer any insight into whether such non-2m HC expression and the vast potential combinatorial dimers play a role in transplantation reactions?
4. The exciting association of B27 FACE-2 expression with ankylosing spondylitis suggests a promising therapeutic target, especially since the HD5 mAb demonstrated an animal proof-of-concept (over a decade ago). Could the authors comment on why such a human mAb therapeutic does not appear in development?
5. It would be quite useful for the authors to express their views on how one might go about utilizing such non-2m HC monomers and dimers as targets for mAb therapies or as antigen-specific immuno-therapeutics for cancer, autoimmunity, and even transplantation. In addition, is there an opportunity for greater accuracy/precisian of diagnoses associated with unique HC alleles, both genetic sequence and protein dimerization?
Author Response
Reply to Rev-2
We sincerely thank the reviewer for complementing the review which is very well organized in structure and thought, especially the helpful "inferences" statement, and for considering that it is quite thorough in both reviewing literature and the authors' genuine interpretations.
COMMENT # 1
- An issue with the manuscript was the excessive formatting errors throughout, especially Greek symbols that challenged the reviewer. In addition, 2 errors in word choice are "grove" instead of "groove", line 58 and "keep" instead of "kept", line 937.
We express regret for the errors. When the journal converted our Word document into another font style, it automatically changed into the error kind. We have corrected all as well as “grove” (in Introduction first para) and “keep” (section 11.2 line 2)
COMMENT # 2
- Due et al demonstrated the "insulin receptor" associated with HLA-I HC molecules, not "insulin", lines 71-75.
We express regret for the error and we have corrected it in the manuscript in Page 2 last para, in other pages as follows:
Page 3:
Cell surface HLA-I molecules are capable of binding to other proteins to exert a signal-transducing function [6, 7]. Due et al. [8] minimized the binding of insulin receptors to the cell surface HLA-I using HC-specific monoclonal antibodies (mAbs) (PA 2.6, and BB 7.5, -6 and-7). However, the B2m-specific polyclonal Abs did not affect such binding of insulin receptors to HLA, which suggests that the cell surface HLA HCs can independently serve as ligands without the involvement of B2m.
Page 20
They can be formed between different sites within the same molecule (intramolecular or intrachain crosslinks) or between two different chains in a single molecule (e.g., the interchain crosslinks in mammalian insulin receptors), and they play a key role in stabilizing or maintaining the structural configurations that are essential to functional activity. The number of S-S bond-containing polypeptide HCs is well defined in structural proteins such as receptors (e.g., the low-density lipoprotein receptor, LDLR), and extracellular matrix proteins (e.g., laminin). Similarly, dityrosine crosslinking is observed in lipoproteins, extracellular matrix proteins, lysozyme, myoglobin, fibronectin, laminins, calmodulin, insulin receptors, hemoglobin, and centrin 2 [161].
And in the other two pages for minor editing.
COMMENT # 3
- Could the authors offer any insight into whether such non-2m HC expression and the vast potential combinatorial dimers play a role in transplantation reactions?
We thank the reviewer for bringing out this aspect.
This area requires very detailed discussion; unfortunately, specific, reliable, and stringent documentation for the expression of both monomers and dimers in immune cells of allograft recipients is lacking, particularly against mismatched alleles;
However, based on the reviewer’s comments, we revisited the manuscript and added a short paragraph (in section 8.2) with new references as follows addressing whether such non-2m HC expression and the vast potential combinatorial dimers play a role in transplantation reactions.
We hope that our hypothesis may stimulate further studies on the multivarious HLA-I antibodies formed against novel HLA-I heterodimers in mismatched organ transplantation that may have a role in transplant rejection. Indeed, proinflammatory cytokines and T cell activation can be expected after recognition of donor HLA antigens, either by the direct pathway (donor HLA on donor antigen-presenting cells), the semi-direct pathway (intact donor HLA on recipients antigen-presenting cells), or by the indirect pathway (self-restricted presentation of processed donor-derived HLA HCs on HLA molecules on recipient antigen-presenting cells) [138]. In this regard, Horuzko’s team of investigators [139-142] has observed HLA-G a high level of HLA-G dimers in circulation and augmented expression of the membrane-bound HLA-G on monocytes are associated with the prolongation of kidney allograft survival. Interestingly, HLA-G dimer in circulation was higher in a group of 90 patients with a functioning allograft compared with 40 patients who underwent allograft rejection. They have demonstrated that the HLA-G dimer inhibits the activation and cytotoxic capabilities of human CD8+ T cells. This mechanism is implicated in the downregulation of Granzyme B expression. Such homo- and heterodimers may function through differential binding to the LILRBI family of receptors [141]. They also documented the presence of soluble HLA-G dimers is associated the lower levels of proinflammatory cytokines, suggesting a major role of HLA-G dimers in controlling inflammatory state.
- Breman, E.; van Miert, P. P. van der Steen, D. M.; Heemskerk, M. H.; Doxiadis, I. I.; Roelen, D.; Claas, F. H.; van Kooten, C. HLA monomers as a tool to monitor indirect allorecognition. Transplantation, 2014, 97, 1119-127. doi: 10.1097/TP.0000000000000113
- Ezeakile M, Portik-Dobos V, Wu J, Horuzsko DD, Kapoor R, Jagadeesan M, Mulloy LL, Horuzsko A.HLA-Gdimers in the prolongation of kidney allograft survival. J Immunol Res. 2014, 2014:153981. doi: 10.1155/2014/153981.
- Ajith A, Portik-Dobos V, Nguyen-Lefebvre AT, Callaway C, Horuzsko DD, Kapoor R, Zayas C, Maenaka K, Mulloy LL, Horuzsko A. HLA-Gdimer targets Granzyme B pathway to prolong human renal allograft survival. FASEB J. 2019, 33, 5220-5236. doi: 10.1096/fj.201802017R.
- Carosella, E. D.; LeMaoult, J. Multimeric structures of HLA-G isoforms function through differential binding to LILRB receptors. Cell Mol Life Sci. 2012, 69, 4041-4049. doi: 10.1007/s00018-012-1069-3.
- Wu, J.; Zhang, W.; Hernandez-Lopez, P.; Fabelo, E.; Parikh, M.; Mulloy, L.L.; Horuzsko A.Isoforms of human leukocyte antigen-Gand their inhibitory receptors in human kidney allograft Hum Immunol. 2009, 70, 988-994. doi: 10.1016/ j.humimm.2009.07.023.
However, in the past 9 years, not many have attempted to focus on the expression of Face-2 and their homo and heterodimers. Keeping this in mind, we have introduced Figure 4 to illustrate possible diversification in donor-derived HLA HC monomers and possible dimerization between donor-derived and recipient HC monomers, and the possible appearance of donor-recipient mixed heterodimers.
However, in the present review, we do not want to elaborate on the immunogenicity of B2m-free HC monomers and heterodimers during allorecognition in allograft recipients. There are two reasons:
Reason 1, this review already nearing the page limit.
Reason 2, we plan to undertake a detailed review following the publication of this manuscript on the dynamics and immunogenicity of b2m-free HLA HCs in normal healthy and allograft recipients, as we have done for the naturally occurring HLA-I antibodies in non-immunized normal and healthy individuals and in allograft recipients. We have some primary publications in the Journal Blood and Clin. Exp. Immunology illustrates this, but not necessarily from the point of allo- and auto HLA monomers and dimers.
COMMENT # 4
- The exciting association of B27 FACE-2 expression with ankylosing spondylitis suggests a promising therapeutic target, especially since the HD5 mAb demonstrated an animal proof-of-concept (over a decade ago). Could the authors comment on why such a human mAb therapeutic does not appear in development?
This specific comment of the reviewer prompted us to research literature on the specificity and functions of mAb HD5. Amazingly the following publication has come to our light and it is strikingly relevant to this review.
- Belaunzaran, M. O.;, Kleber, S.; Schauer, S.; Hausmann, M.; Nicholls, F.; Van den Broek, M.; Payeli, S.; Ciurea, A.; Milling, S.; Stenner, F.; Shaw, J.; Kollnberger, S.; Bowness, P.; Petrausch, U.; Renner, C. HLA-B27-Homodimer-Specific AntibodyModulates the Expansion of Pro-Inflammatory T-Cells in HLA-B27 Transgenic Rats. PLoS One. 2015, 10(6):e0130811. doi: 10.1371/journal.pone.0130811.
Therefore we have taken the liberty to revise this aspect in our review as follows in Section 7 as follows:
Most importantly, Belaunzaran et al [123] elucidated the functional role of B27-homodimers (Face-3 of B27) using a B27-homodimer-specific mAb (HD5). The mAb HD5 bound specifically to B27-homodimers but not b2m-associated intact HLA-B27 or A1, B7, B13, or C7. The B27-Face-3 binds to members of the killer cell receptor (KIR) and leukocyte Ig-like families (details vide infra). In addition, in vitro assays demonstrated that B27-Face-3 interactions with CD4+ T cells induced proinflammatory responses (secretion of TNF and IFN-g). Therefore, it was postulated that mAb HD5 may block the interaction of B27-Face-3 to alter the immune responses. Indeed, mAb45 resulted in reduced soluble TNF and decreased CD4+ T cells. This investigation of HD5 mAb necessitates the need for developing therapeutic human HD5-specific monoclonal antibodies for treating spondyloarthropathies. Furthermore investigating serum antibodies against B27-Face-3 in spondyloarthropathies may yield more information on the pathogenesis and progression of the disease and enable the development of other specific and personalized therapies.
COMMENT # 5
- It would be quite useful for the authors to express their views on how one might go about utilizing such non-2m HC monomers and dimers as targets for mAb therapies or as antigen-specific immuno-therapeutics for cancer, autoimmunity, and even transplantation. In addition, is there an opportunity for greater accuracy/precisian of diagnoses associated with unique HC alleles, both genetic sequence and protein dimerization?
We thank the reviewer for this wonderful suggestion. In fact, keeping this concern in mind, we have introduced the controversial Figure 4. The very purpose of Figure 4, is to prompt investigators to study the immunology of HLA HC mono, homo, and heterodimers for developing vaccine-based active specific or antibody-based passive specific immunotherapy against cancer, autoimmunity, end-stage organ diseases, and in transplantation. In light of these comments, we revised the section involving Figure 4 as follows (red new, black existing version).
Not only HLA typing must be known of patients for developing personalized immunotherapy, but consideration of the immunogenicity and antigenicity of homo- and heterodimers is also potentially critical. Based on Table 4, we visualize approximately 144 monoclonal antibodies specific for each one of these dimers. Similarly, the patients’ sera have to be screened against these monomers and dimers and most importantly against the dimers because they interact with both KIR and leukocyte Ig-like family of receptors. We hope that our hypothesis may stimulate further studies on the multivarious HLA-I antibodies formed against novel HLA-I heterodimers in mismatched organ transplantation that may have a role in transplant rejection. Similarly, the possible role of these dimers in lymph node and organ metastasis of human cancer as well as in autoimmune diseases needs to be investigated for developing personalized active specific and passive immunotherapies.
We have also extensively revised the Figure 4 legend, as it is very important for future investigations
Figure 4. Diagrammatic illustration of different combinations of dimers of peptide-free (implying unfolded) monomeric B2m-Free HCs (Face-2) resulting in different kinds of peptide-free (implying unfolded) allelic homodimers (Face-3), peptide-free (implying unfolded) allelic heterodimers (Face-4A) and peptide-free (implying unfolded) isomeric heterodimers (Face-4B). For the purpose of illustrating a similarity with HLA-II, we have suggestively and tentatively indicated one monomer as a chain and another monomer as a beta chain, although in reality both of them are alpha chains. As defined and designated by IMGT information systems, each HC, though considered unfolded, may consist of the groove (G)-like domains (a1 [D1] and a2 [D2]) and a constant ©-like domain (a3 [D3]) with a cytoplasmic tail. Domains of different alleles of different isomers or the a and b chains as shown in the figure may vary extensively. It implies that the ligands binding to the alpha1 domain of these two chains may differ markedly. More structural studies on the dimers would further clarify the G and C domains of each chain in a homodimer and in the allelic and isomeric heterodimers.
We once again thank the reviewer profusely for improving the manuscript significantly. We are much obliged for the same.

Reviewer 3 Report
The Ravindranath first and second authors and their team have presented an interesting and comprehensive, even historical, review of b2m-free MHC-1 heavy chains (HC) that may reach cell surface expression in monomeric and/or multimeric (such as dimeric) form. I felt that published data was described and interpreted in a fair and intelligent way, and I was pleased to read this work on a topic that can generate controversy, when one wonders if perhaps its details should be more accepted in the conventional immunological canon. For example, we talk about mice without b2m and call them mice with, therefore, no MHC-1, even when we add b2m and exogenous peptide knowing that b2m-free HCs are at the surface and thus load for T cell stimulation. Yet we still discuss the APCs as if they have no MHC-1. This review faces that contradiction directly, clarifies it with data and interpretation, and is thus very timely and may correct a generalized fallacy that is common in the field. I am quite in favor of this article, although I will bring up some issues that could be resolved, below.
(1) Line 111 makes a claim where I do not know of data supporting that. I think that after antigen presentation to T cells, the peptide does not fall out of the groove, but rather single peptide-MHC complexes can engage multiple TCRs and T cells. Then lines 112-113 make another claim, with reference 9, but I do not believe reference 9 supports those points.
(2) Lines 152-155, and preceding supporting text, and also later text stating that b2m-free HCs have different glycosylation pattern (lacking terminal sialic acid caps on glycans) compared with conventional b2m+HC MHCs. This could be a marker of a change affecting many glycoproteins and glycolipids in cells, and one wonders if this might not be MHC-specific. Perhaps it is happening to folding-failed HC, but also to many other molecules at once?
(3) Like above, stimulation with IFN, PMA, or other stimulants are stated as causing increase in b2m-free HC expression. Is there evidence that this is MHC-specific? One could think that many molecules could be affected by a common stimulation-induced process, and perhaps there is not specificity for MHC, but rather many folding-failed proteins are escaping conventional quality control mechanisms that would otherwise degrade them.
(4) Somewhat related to the point above, how well do we know that b2m-free HC monomers are monomers, and are not physically associated with something else? Most flow cytometry stains and also Western blots will have looked for MHC-related proteins. But if stimulated cells are now failing to degrade many (MHC and other) proteins that fail to finish conventional folding, and they do not mature normally through Golgi secretory pathway (thus presenting immature glycans), might "monomeric" HCs be tied in knots (meaning, physically associated in irregular ways) with other proteins all going through the same expression pathway? Only sometimes is the binding partner another MHC-related protein? Using a mAb specific for b2m-free HC to immunoprecipitate and then look for co-associated proteins could perhaps attempt to clarify if b2m-free HC "monomers" are truly "alone", but when dealing with membrane-spanning or -associated proteins the proteomics could be challenging, too. The issue is brought up in this critique in case the authors know of clarifying data if it exists, so we could be confident that b2m-free HC "monomer" is for sure a monomer.
(5) "Dimer" is also a very precise term. It is understood here to refer to two proteins (not a larger number) mainly because of S-S bonds often at parallel sites in homogenous binding partner that would covalently join two together. But the b2m interaction with HC is non-covalent, another type of dimer. The issue perhaps comes best into focus with data cited about two different HCs forming different size clusters, which makes it sound like "dimers" may not be dimers but may form larger complexes (around line 489). To summarize the question, are we sure b2m-free HC "dimers" are dimers, or could they be larger complexes with >2 proteins in them? Would such complexes be specific for MHC-related proteins, or would MHC-non-specific molecules be present too?
(6) The authors have done an outstanding job regarding mAb specificity, previous beliefs about that, and later illuminating updates correcting errors. It does make it difficult for the reader to read about W6/32, however. After the authors update that it can bind both b2m-free and -bound HCs, later they still refer to data interpreting it the original way. Is it possible to discuss all data, balanced less on its historical interpretation, and more in light of the specificities we now understand to be more correct? I understand the authors are describing historical interpretation as well as updated modern, so choices made regarding this I leave to discretion of the authors and editors.
(7) Line 429, what is meant by "irreversible"?
(8) This reviewer's copy of the manuscript, both online and in PDF format, failed to show greek letters for subunits beta, alpha, gamma, etc. I knew what was meant being familiar with the field, but wanted the authors and editors to know this and correct it. Also, Table 3 went off the page to the right and that part could not be viewed.
(9) All of section 11 seemed to be a repetition of the previous parts of the paper. Perhaps bring out the novel arguments (even if supported by previously explained data) and make more succinct. But I leave the choice of how to approach this to the discretion of the authors and editors.
Again I express positivity regarding the authors' thoughtful review, interpretation, and expression of provocative hypotheses.
Author Response
Reply to Rev-3
COMMENT # 1
Line 111 makes a claim where I do not know of data supporting that. I think that after antigen presentation to T cells, the peptide does not fall out of the groove, but rather single peptide-MHC complexes can engage multiple TCRs and T cells. Then lines 112-113 make another claim, with reference 9, but I do not believe reference 9 supports those points.
REPLY:
- We agree with the reviewer that the Reference [9] is incorrect.
- Furthermore, based on the concern of the reviewer that “the peptide does not fall out of the groove,” after antigen presentation, we surveyed the literature again in depth.
- The literature documents that the peptide does fall out of the groove. And it happens as soon as B2m is dissociated (we have provided details below with figures and references).
- However, to support the claim made in the figure, I wish to present first, the schematic figures provided by two different investigators in different publications in 1992 and 1999. The diagrams justify figure 2, although there is no clear-cut evidence that it happens after antigen presentation in these two papers.
FIRST EVIDENCE is presented in Figure 1 of the following paper
Elliott T. How do peptides associate with MHC class I molecules? Immunol Today. 1991 Nov;12(11):386-8. doi: 10.1016/0167-5699(91)90134-f.
This figure is redrawn to clarify the legends (A long peptide is Unstable, Short peptide is stable)
”the affinity of heavy chains for peptide increases dramatically in the presence of b2m. That is, since K3>K1 and K2>K4, the binding of peptides and [b2-m to heavy chains is cooperative, and peptide binding and class I assembly are, therefore, linked phenomena”
[Immunology Today. 1991, 12 (11): page 386 middle column, upper paragraph]
Elliott T, Cerundolo V, Elvin J, Townsend A. Peptide-induced conformational change of the class I heavy chain. Nature. 1991 May 30;351(6325):402-6. doi: 10.1038/351402a0.
“Specific short peptides (9–10 amino acids) can induce folding of the heavy chain in the absence of β2m. Both short (nine amino acids) and longer sequences (15 amino acids) can stabilize preformed low-affinity complexes of heavy chain and β2m.”
Cerundolo V, Elliott T, Elvin J, Bastin J, Rammensee HG, Townsend A. The binding affinity and dissociation rates of peptides for class I major histocompatibility complex molecules. Eur J Immunol. 1991,21(9):2069-75. doi: 10.1002/eji.1830210915.
“Our results suggest that although longer peptide sequences may bind in the endoplasmic reticulum, they are likely to have dissociated by the time the class I molecules arrive at the cell surface. Such class I-B2m complexes would then be expected to dissociate [16, 28]. Such a mechanism could contribute to the pool of free heavy chains that have been detected at the surface of non-mutant cells ([58-601 and H. Ploegh and N. Stam PhD Thesis, Amsterdam; personal communication).”
“Even short peptides (9 amino acids) “form stable complexes with half-lives (of b2m-associated HLA) greater than 110 h at 4 degrees C, 39 h at 22 degrees C, and 3 h at 37 degrees C.”
Therefore, at the termination of the functions of the intact HLA molecules (Face-1), B2m-dissociates and leaves HCs to be cleaved by MMP.
Further supporting evidence is as follows: SECOND EVIDENCE is presented in Figure 5 of the following paper
Demaria S, DeVito-Haynes LD, Salter RD, Burlingham WJ, Bushkin Y. Peptide-conformed beta2m-free class I heavy chains are intermediates in generation of soluble HLA by the membrane-bound metalloproteinase. Hum Immunol. 1999 60(12):1216-26.
FIGURE 5. Generation of conformed and non-conformed soluble β2m-free class I HC. Dissociation of HC/β2m complexes with bound peptides (Ag) on the cell surface results in the generation of two membrane forms, conformed HC with bound peptides (upper) and non-conformed HC without peptides (lower). Both membrane forms of free HC are released into supernatants by a membrane-bound MPase. The soluble HC released by this enzyme may have different fates. Conformed HC can reassociate with β2m in solution and retain their conformations or, alternatively, lose their peptides and become non-conformed HC. Non-conformed HC is likely to be degraded in solution.
H. Ploegh and N. Stam PhD Thesis, Amsterdam; personal communication).”
“Even short peptides (9 amino acids) “form stable complexes with half-lives (of b2m-associated HLA) greater than 110 h at 4 degrees C, 39 h at 22 degrees C, and 3 h at 37 degrees C.”
Therefore, at the termination of the functions of the intact HLA molecules (Face-1), B2m-dissociates and leaves HCs to be cleaved by MMP.
Further supporting evidence is as follows: SECOND EVIDENCE is presented in Figure 5 of the following paper
Demaria S, DeVito-Haynes LD, Salter RD, Burlingham WJ, Bushkin Y. Peptide-conformed beta2m-free class I heavy chains are intermediates in generation of soluble HLA by the membrane-bound metalloproteinase. Hum Immunol. 1999 60(12):1216-26.
FIGURE 5. Generation of conformed and non-conformed soluble β2m-free class I HC. Dissociation of HC/β2m complexes with bound peptides (Ag) on the cell surface results in the generation of two membrane forms, conformed HC with bound peptides (upper) and non-conformed HC without peptides (lower). Both membrane forms of free HC are released into supernatants by a membrane-bound MPase. The soluble HC released by this enzyme may have different fates. Conformed HC can reassociate with β2m in solution and retain their conformations or, alternatively, lose their peptides and become non-conformed HC. Non-conformed HC is likely to be degraded in solution.
However, we wish to bring to the attention of the reviewer the following facts for reconsideration of his comments. We will revise the paragraph under this section with new references cited.
Furthermore, we bring to the kind attention of the reviewer the following references too, which further substantiate the findings presented above.
- Demaria S, DeVito-Haynes LD, Salter RD, Burlingham WJ, Bushkin Y. Peptide-conformed beta2m-free class I heavy chains are intermediates in generation of soluble HLAby the membrane-bound metalloproteinase. Hum Immunol. 1999, 60,:1216-1226. doi: 10.1016/s0198-8859(99)00113-5.) and
- DeVito-Haynes LD, Demaria S, Bushkin Y, Burlingham WJ. The metalloproteinase-mediated pathway is essential for generation of soluble HLAclass I proteins by activated cells in vitro: proposed mechanism for soluble HLA release in transplant rejection. Hum Immunol. 1998, 59, 426-434. doi: 10.1016/s0198-8859(98)00032-9.)
Based on these papers, we have summarized the two pathways for the formation of B2m-free HCs (Face-2).
# 1: The over-expression of Face-2, induced by PMA, was examined using BFA, which arrests the exit of newly synthesized proteins from the ER. Demaria et al. state “Expression of these molecules was completely inhibited by BFA and thus induction of B2m-free class I heavy chains on activated T cells requires the transport of newly synthesized MHC class I molecules to the surface”(page 107, lines 5-8).
#2: In the second experiment, they stripped Face-2 from PMA-activated T cells by briefly treating them with trypsin in the cold. They state that “treatment of these cells with trypsin selectively removed surface B2m-free class I heavy chains but had no effect on B2m-associated MHC class I molecules.” (p.107, Lines 19-21). The treated cells were then incubated at 37oC in the presence or absence of BFA. They state that “BFA did not block their reappearance on these trypsin-treated activated T cells. Thus, it appears that at least some B2m-associated MHC class I molecules on the surface of activated
T cells can dissociate and give rise to B2m-free class I heavy chains.” They are immunostained with mAb HC10 to confirm the emergence of Face-2 due to the dissociation of B2m from Face-1.
They further carried out radiolabeling and immunoprecipitation experiments on these cells with mAbs (HC-10 & BB7.7). The results reveal that the surface-expressed Face-1 molecules “did not alter the quantity available to immunoprecipitation with Sepharose-Protien A beads” (p. 110) but Face-2 molecules “were clearly absent in the immunoprecipitates. They state that “this “disappearance” of b2m-free class I heavy chains could be due to internalization…. or due to “their release into the medium”. In this paper, they even hypothesize “the proteolytic cleavage of surface b2m-free class I heavy chains may be involved.”
This hypothesis was further revalidated in their next paper (Demaria et al 1999).
Based on the comment of the reviewer, we have extensively revised the paragraph and the Figure with more supporting references as shown below
2.1. Are cell surface B2m-free HCs ephemeral due to dissociation of B2m?
In Face-1 HLA trimers, an association of B2m with HC increases the affinity of HC for peptides. Similarly, B2m increases the stability of the peptide on the groove of the HC [11-17]. In a unique experiment, Demaria et al [11] selectively removed any B2m-free HCs (Face-2) on the cell surface of PMA-activated T cells with a brief treatment of cold trypsin. This treatment did not remove B2m-associated HCs on the cell surface. They exposed the PMA-treated resting T cells and trypsin-treated PMA-activated cells to BFA, which arrests the exit of newly synthesized proteins from the ER. Interestingly, the BFA blocked the expression of Face-2 on resting T cells exposed to PMA but did not block their reappearance of trypsin-treated cells, documenting the dissociation of B2m from intact Face-1 molecules. The dissociation of B2m results in the unfolding of alpha1 and alpha 2 domains of HCs, these are referred to as “non-conformed” B2m-free HCs. These “non-conformed” B2m-free HCs are quite distinct from folded a1 and a2 domain bearing HCs or “conformed HCs, as is observed on cells normally expressing functional B2m [17]. Notably, such peptide-carrying conformed B2m-free HCs are also observed in B2m-deficient mice [17]. Figure 2 illustrates the dissociation of B2m from the B2m-associated intact HLA (Face-1), the consequent release of peptides, and the formation of non-conformed membrane-bound HCs. which are selectively cleaved by membrane metalloproteinases (MMPs) [12-14]. Hence the cell surface non-conformed B2m-free HCs are considered to have a very insignificant half-life. Figure 2 is based on the findings and illustrations of Elliott [14] and Demaria [12].
Figure 2. Events occurring after dissociation of B2m, based on the previous reports [11-17]. It has been well documented [16] that the human mutant cell T2 expressing H-2Db HCs, carrying 9 amino acid peptide, complexed with human B2m form stable complexes with half-life > 110h at 4oC, 39h at 22oC and 3h at 379C. They have also shown that small increases in the length of the peptide greatly reduce the half-life (t ½ to about 1-10 h at 4oC). Transformation of conformed HC to Non-conformed HC and cleavage of both conformed and non-conformed HC by MMPs and proteolytic degradation of B2m and HC [11, 12, 14]. Proteolytic modifications of B2m in sera are based on the works of Nissen [18, 19].
The proposition that HLA-I HCs cannot be expressed on the cell surface without B2m is based on the unique properties of two cell lines. The first is Daudi, the Burkitt Lymphoma cell line. It lacks the ability to synthesize B2m and cannot express trimeric HLA-I molecules (Face-1) on the cell surface [20]. The surface expression of Face-1 in the cell line was achieved by supplementing either with mouse or human B2m. The second is the R1 cell line of a somatic cell variant of the C3H (H-2k) thymoma [21].
COMMENT # 2
(2) Lines 152-155, and preceding supporting text, and also later text stating that b2m-free HCs have different glycosylation pattern (lacking terminal sialic acid caps on glycans) compared with conventional b2m+HC MHCs. This could be a marker of a change affecting many glycoproteins and glycolipids in cells, and one wonders if this might not be MHC-specific. Perhaps it is happening to folding-failed HC, but also to many other molecules at once?
While we appreciate the comment of the reviewer. Of course, Marinko et al (2019, cited below) discuss how glycosylation pattern may be altered during misfolding of human membrane proteins, without any reference to HLA. However, no HLA investigators who have observed differences in glycosylation patterns between B2m associated and free HCs recognize this as a common feature of other cell surface molecules. Moreover, HLA monomers are not misfolded versions but unfolded versions due lack of B2m. Possibly glycosylation of HCs may occur before unfolding of a1 and a2 domains and such altered glycosylation may impact folding! These speculations do remain to be elucidated in greater depth. Now we focus only on the differences observed between B2m-free HCs versus B2m-associated HCs. The differences observed are not consistent as some variation in the density of mannose residues and others with sialic acid; again with sialic acid (NeuAc) whether there is any difference in the linkages such as 2,3; 2,6 or even 2,8 associated with disialyl residues. This needs to be studied in greater depth, but it is beyond the scope of the review. We need to tabulate all publications involving glycosylation of HLA monomer, homo- and heterodimer, and B2m-positive heterodimers.
Marinko JT, Huang H, Penn WD, Capra JA, Schlebach JP, Sanders CR. Folding and Misfolding of Human Membrane Proteins in Health and Disease: From Single Molecules to Cellular Proteostasis. Chem Rev. 2019 May 8;119(9):5537-5606. doi: 10.1021/acs.chemrev.8b00532.
The personal experience of the first author as Director of Glycoimmunotherapy at UCLA & John Wayne Cancer Institute under the leadership of Late Prof. Donald Lee Morton has extensive experience in the glycosylation patterns of different gangliosides such as GM1, GM2, GM3, GD2, GD3, GD1a, O-Ac-GD2, etc. Glycosylation within a tumor cell is found to be specific for each ganglioside. For example, when 9-O-Ac glycosylation occurs on NeuAc2,8NeuAc is specific for the ganglioside but will not occur even on NeuAc2,3Gal or NeuAc2.6Gal. (see publications of Ravindranath M.H. in pubmed.ncbi.nih.gov) Therefore, I infer that glycosylation could be specific for HLA molecules.
COMMENT # 3
Like above, stimulation with IFN, PMA, or other stimulants are stated as causing increase in B2m-free HC expression. Is there evidence that this is MHC-specific? One could think that many molecules could be affected by a common stimulation-induced process, and perhaps there is not specific for MHC, but rather many folding-failed proteins are escaping conventional quality control mechanisms that would otherwise degrade them.
Although this is a thought-provoking suggestion, at present this cannot be ascertained. However, it should be noted that an increase in B2m-free HC expression is evident in many human cancers (e.g. Table 2), during inflammation and infections. In human cancer, B2m-associated HLA class I molecules are distinctly downregulated, when B2m-free HC expression is enhanced. Interestingly such downregulation is loci-specific and varies in different cancers, as pointed out by Garrido (25), in a book entitled “MHC class-I Loss and cancer immune escape”.
Therefore stimulation with IFN, PMA, and other stimulation may act at specific gene levels, and they likely impact HLA genes more specifically than others. However, a critical review is needed to address the comment with evidence, pro or against. We thank the reviewer for the thought-provoking suggestion.
COMMENT # 4
Somewhat related to the point above, how well do we know that b2m-free HC monomers are monomers, and are not physically associated with something else? Most flow cytometry stains and also Western blots will have looked for MHC-related proteins. But if stimulated cells are now failing to degrade many (MHC and other) proteins that fail to finish conventional folding, and they do not mature normally through Golgi secretory pathway (thus presenting immature glycans), might "monomeric" HCs be tied in knots (meaning, physically associated in irregular ways) with other proteins all going through the same expression pathway? Only sometimes is the binding partner another MHC-related protein? Using a mAb specific for b2m-free HC to immunoprecipitate and then look for co-associated proteins could perhaps attempt to clarify if b2m-free HC "monomers" are truly "alone", but when dealing with membrane-spanning or -associated proteins the proteomics could be challenging, too. The issue is brought up in this critique in case the authors know of clarifying data if it exists, so we could be confident that b2m-free HC "monomer" is for sure a monomer.
We beg to differ from the reviewer’s contention as we read almost all papers critically on the issue of monomerization. Several authors have used monomer-specific monoclonal antibodies. Several others (e.g. HD5) and we have developed monomer-specific monoclonal antibodies (TFL-006, patented), using the most commonly shared HLA-Ia and HLA-Ib peptide sequences that remain cryptic (AYDGKDY & LNEDLRSWTA) due to the folded state in the presence of B2m and exposed when they are devoid of B2m or unfolded (open conformers) (see Ravindranath et al in references cited in the manuscript).
Furthermore, several authors have used HC10 and other mAbs for identifying the monomers both on the cell surface and on solid matrices.
Most importantly, Demaria and co-investigators (see reply to Concern 1) have done invaluable experiments to document true monomers, both folded and unfolded. We have already narrated in reply to Concern 1. We have also reiterated in the newly revised section 11 Summary as follows
Since finding HLA-B8 and B27 B2m-free HCs (Face-2) on the cell surface of a human lymphoblastoid cell line [22], their genesis has remained controversial. However, the experimental investigations of Demaria et al [11-13] clarified that Face-2 may emanate from two distinct pathways:
- a) Inducing the expression of B2m-free HCs with PMA on activated T cells and blocking their exit with BFA from the ER together confirms that the induction of Face-2 on activated cells requires direct transport via ER independent of their association with B2m.
- b) In a second experiment, they stripped Face-2 from PMA-activated T cells by brief treatment with trypsin in the cold. Treatment of these cells with trypsin selectively removed surface B2m-free HCs but had no effect on B2m-associated HLA (Face-1) (p.107, Lines 19-21 [11]). The treated cells were then incubated at 37oC in the presence or absence of BFA. They noted that the BFA did not block Face-2 reappearance on these trypsin-treated activated T cells, indicating that they are dissociated from B2m-associated HLAs. They were immunostained with mAb HC10 to confirm the emergence of Face-2 is due to the dissociation of B2m from Face-1.
COMMENT # 5
"Dimer" is also a very precise term. It is understood here to refer to two proteins (not a larger number) mainly because of S-S bonds often at parallel sites in homogenous binding partner that would covalently join two together. But the b2m interaction with HC is non-covalent, another type of dimer. The issue perhaps comes best into focus with data cited about two different HCs forming different size clusters, which makes it sound like "dimers" may not be dimers but may form larger complexes (around line 489). To summarize the question, are we sure b2m-free HC "dimers" are dimers, or could they be larger complexes with >2 proteins in them? Would such complexes be specific for MHC-related proteins, or would MHC-non-specific molecules be present too?
Again a thought-provoking comment and we thank the reviewer for the same.
Dimerization is well documented in the literature. Many believed that they may be joined covalently. That is a logical interpretation, as a1 and a2 domains remain unfolded in both HCs of a dimer. That is the main reason we have written this primary section of the article in chronological order of references.
Taking into consideration the comment of the reviewer, we have added a new paragraph in section II Summary as follows with suitable but restricted references, clearly documenting dimers.
‘The half-life of B2m-free HCs (Face-2) seems to depend on peptide binding to the HCs [94] or on clustering and dimerization with other Face-2 molecules. While constituting liposomes, it was noted that Face-2 self-associated with other Face-2 molecules [121]. Similarly, Face-2 molecules cluster on activated normal B and T cells, on cells of B and T lymphoblast lines, and on transformed fibroblasts [122-126]. No such clustering was observed on the surfaces of resting B or T cells or normal fibroblasts. Interestingly, the clustering was reversed by exogenous B2m. Novel homo (Face-3) and hetero dimers (Face-4) of Face-2 molecules are observed on the exosomes released from human monocyte-derived dendritic cells [131] and on different cell lines [132, 133]. Dimerization of Face-2 molecules of HLA-G [134] and HLA-F [135] is also observed.’
One of the most senior and highly respected HLA Scientists is Jack Strominger and a colleague of my supervisor, Later Prof. Paul I Terasaki. We respect the works of Strominger highly. He has studied dimers in greater depth. We have narrated his teams work in section 8.4 as follows:
“Boyson in Schrominger’s group [152] investigated whether HLA-G dimerization occurred naturally, using the HLA-G-transfected mutant EBV-transformed B cell line B-LCL 721.221 which is devoid of any HLA class I molecules. The B-LCLs were lysed and analyzed on SDS/PAGE gels under both reducing and non-reducing conditions, blotted to nitrocellulose. On the blots, the HLA-G HCs were monitored with the HLA-G HC-specific mAb MEM-G/1. Under non-reducing conditions, two bands, one 39 kDa and another 78 kDa were observed. However, under reducing conditions, the 78kDa band was not observed, suggesting that the band missing in reducing gels was a dimer, possibly disulfide-bonded. Further confirmation of the dimeric nature of the 78 kDa band was ascertained by a parallel control experiment. They repeated the experiments done by another Strominger team [153] in which HLA-A2 was immunoprecipitated by using anti-β2m mAb (BBM.1) beads from surface-biotinylated HLA-A2 transfectants in the presence and absence of iodoacetamide (IAM). Dimerized HLA-A2 HCs were observed under nonreducing conditions in the absence of IAM, but this dimerization was completely abrogated by the addition of IAM, indicating that the HLA-A2 dimers could be an artifact of cell lysis and immunoprecipitation. The inclusion of IAM in the lysis buffer enabled the distinguishing between preexisting and artefactual HLA dimers. Similarly, when HLA-G was immunoprecipitated from surface-biotinylated HLA-G transfectants, HLA-G dimers were detected under nonreducing conditions even in the presence of IAM, suggesting that they were preexisting dimers and not an artifact. The existence of dimers was confirmed using the conformation-specific W6/32 mAb and the HLA-G-specific MEM-G/11 mAb. Furthermore, the mutation of C42 on HLA-G to Ser42 completely abrogated dimerization of HLA-G in the absence of IAM. Thus, dimerized HLA-G exists on the cell surface linked by means of a C42-mediated disulfide bond. Gonen-Gross et al. [99, 100] documented the involvement of C42 in the dimerization of HLA-G using the B cell line 721.221 and a melanoma cell line LB33 mel B1 that expresses HLA-A24 only. Similarly, Shiroishi et al. [154] showed that HLA-G can be expressed as a disulfide-linked dimer both in solution and at the cell surface.”
Taking into consideration of the reviewer’s comment, we revisited the literature on the reliability of B2m-free HC dimers. In section 8.2, we have added the following evidence to document that dimers are indeed dimers but not just clusters.
“In this regard, Horuzko’s team of investigators [139-142] have observed HLA-G a high level of HLA-G dimers in circulation and augmented expression of the membrane-bound HLA-G on monocytes are associated with prolongation of kidney allograft survival. Interestingly, HLA-G dimer in circulation was higher in a group of 90 patients with a functioning allograft compared with 40 patients who underwent allograft rejection. They have demonstrated that the HLA-G dimer inhibits the activation and cytotoxic capabilities of human CD8+ T cells. This mechanism is implicated in the downregulation of Granzyme B expression. Such homo- and heterodimers may function through differential binding to the LILRBI family of receptors [141].”
- Ezeakile M, Portik-Dobos V, Wu J, Horuzsko DD, Kapoor R, Jagadeesan M, Mulloy LL, Horuzsko A.HLA-Gdimers in the prolongation of kidney allograft survival. J Immunol Res. 2014, 2014:153981. doi: 10.1155/2014/153981.
- Ajith A, Portik-Dobos V, Nguyen-Lefebvre AT, Callaway C, Horuzsko DD, Kapoor R, Zayas C, Maenaka K, Mulloy LL, Horuzsko A. HLA-G dimer targets Granzyme B pathway to prolong human renal allograft survival. FASEB J. 2019, 33, 5220-5236. doi: 10.1096/fj.201802017R.
- HoWangYin, K. Y.; Loustau, M.; Wu, J.; Alegre, E.; Daouya, M.; Caumartin, J.; Sousa, S.; Horuzsko, A.; Carosella, E. D.; LeMaoult, J. Multimeric structures of HLA-G isoforms function through differential binding to LILRB receptors. Cell Mol Life Sci. 2012, 69, 4041-4049. doi: 10.1007/s00018-012-1069-3.
The dimers investigators can be classified tentatively into four groups.
Group 1. Investigators dealing with non-specific dimers of the same alleles of a locus.
Group 2. Then Section 7 of the review on authors dealing with B27 in spondyloarthropathies. They have carried out a detailed investigation on S-S bonding between HCs at Cysteine 67.
Group 3. Based on these observations, other investigators observed s-s bonding between heterodimers (HLA-B and HLA-C & HLA-C and HLA-F). In HLA-C and HLA-A cysteine is absent at position 67 but observed at other positions (see Tables 3-5).
Group 4. We have shown the presence of Tyrosine at many positions in Table 3-5 and brought to the attention of the readers that there is evidence in the literature on other proteins on the role of tyrosine in the formation of dityrosine linkage. Although plenty of tyrosine is present in all HLA-I HCs, there are no studies done on HLA molecules to examine the occurrence of dityrosine linkages in HCs. Possibly some readers who have studied this review may test the hypothesis and bring this to light.
Indeed, H-H bonding, Salt linkages, S-S bonding, and possible dityrosination may be found in the homo and heterodimers. More studies are needed not only for this purpose on HCs but as the reviewer pointed out monomeric HC linkage with other proteins needs to be studied. We may have to wait for ten or more years to see the proof better.
COMMENT # 6
The authors have done an outstanding job regarding mAb specificity, previous beliefs about that, and later illuminating updates correcting errors. It does make it difficult for the reader to read about W6/32, however. After the authors update that it can bind both b2m-free and -bound HCs, later they still refer to data interpreting it the original way. Is it possible to discuss all data, balanced less on its historical interpretation, and more in light of the specificities we now understand to be more correct? I understand the authors are describing historical interpretation as well as updated modern, so choices made regarding this I leave to discretion of the authors and editors.
We are happy to note our concerns about W6/32 based on Trans work.
As stated in section 8.1
Although mAb W6/32 was used with the notion that it may stain only intact B2m-associated HLA molecules, the work of Tran et al. [86] showed that the mAb W6/32 “actually recognizes an epitope present on isolated non-reduced α-chains of most HLA-B allelic forms.”
Compare W6/32 binding on Luminex beadsets coated only with B2m-associated HCs and only with B2m-free HCs. As we have shown in our earlier publications
- Ravindranath, M.H.; Jucaud, V.; Ferrone, S. Monitoring native HLA-I trimer specific antibodies in Luminex multiplex single antigen bead assay: Evaluation of beadsets from different manufacturers. Immunol. Methods, 2017, 450, 73–80. doi: 10.1016/j.jim.2017.07.016.
Jucaud, V.; Ravindranath, M.H.; Terasaki, P.I. Conformational Variants of the Individual HLA-I Antigens on Luminex Single Antigen Beads Used in Monitoring HLA Antibodies: Problems and Solutions. Transplantation 2017, 101, 764–777.
- Only the beadsets obtained from Immucor contain only B2m-associated HCs but not the beadsets from One Lambda Inc, which carried both B2m-associated and B2m-free HCs.
- Critical evidence needed for confirming the specificity of a monoclonal antibody is the inhibition of W6/32 binding to B2m-associated or B2m-free HCs by inhibiting the binding with B2m-free HCs. Trans observations are good but need further confirmation. The same is true for the other argument that W6/32 is specific for B2m-associated HCs. Unless this is done systematically and critically, the antibody affinity of W6/32 is speculative and not confirmative.
COMMENT # 7
Line 429, what is meant by "irreversible"?
Vasquez et al. [113] studied two untransfected cell lines, Hmy2.CIR and a human lymphoid cell line T2, after transfecting with various B27 alleles (B*27:04, B*27:05, B*27:06, and B*27:09). Flow cytometric observations using an intact (Face-1) HLA-B27 specific mAb ME1 (IgG1) and an HC-specific mAb HC10 revealed that irreversible forms ofï€ Face-2 appeared at the cell surface to a similar extent among all subtypes, irrespective of their association with Ankylosing Spondylitis.
Vasquez et al argue that the monomers of T2 transfected with various alleles reacted identically with both the mAbs, one specific for B27 and another specific for HC-10 to confirm that there are no other monomers different from B27 positive ones. We retained the term “irreversible” to keep the sentiments of the authors to highlight the fact that no other monomers different from B27 positive ones occur in these two untransfected cell lines transfected with B27 alleles. The cell lines did not have any other monomers other than B27.
COMMENT # 8
This reviewer's copy of the manuscript, both online and in PDF format, failed to show greek letters for subunits beta, alpha, gamma, etc. I knew what was meant being familiar with the field, but wanted the authors and editors to know this and correct it. Also, Table 3 went off the page to the right and that part could not be viewed.
In the revised manuscript, the Corresponding author takes all efforts to correct the Greek letters.
In the revised manuscript, the Corresponding author already corrected table 3 and we will leave a specific note to publishers about our serious concern and plan to suggest placing the table horizontally on the page.
COMMENT # 9
All of section 11 seemed to be a repetition of the previous parts of the paper. Perhaps bring out the novel arguments (even if supported by previously explained data) and make more succinct. But I leave the choice of how to approach this to the discretion of the authors and editors.
Based on the comment of the reviewer, we have not only condensed section 11, but also removed the subheading and combined section 12. Hope this is satisfactory to the reviewer. We need to be redundant sometimes in order to emphasize some points of view, more particularly in this extensive review.
We wish to thank the Reviewer once again for his or her kind compliments and appreciation of the work. We are so much delighted that the reviewer has gone line by line into each and every aspect presented in the manuscript. We hope and pray that our readers of this review do the same so that the hypotheses are not abandoned but continue to stimulate research to prove for or against to evolve the concept.

Round 2
Reviewer 3 Report
The authors have addressed my questions, and I appreciate their thorough treatment even when some questions required speculation and merit future experimentation.
Author Response
We thank the reviewer for critically going through the revised manuscript and for offering valuable comments to improve the manuscript further
Reply to Reviewer # 3
Thank you for your note stating
“The authors have addressed my questions, and I appreciate their thorough treatment even “when some questions required speculation and merit future experimentation.”
We have gone through your question carefully and decided to add our reply as a concluding remark or last paragraph of the last section as follows:
In conclusion, this review illustrates how different variants of HLA molecules emerge on the cell surface, independent of the B2M-associated HLA that we have designated as Face-1. Experimental investigations have demonstrated that variants (i.e., B2M-free monomers that we have designated as Face-2) can traverse from the ER to the cell surface independent of Face-1, particularly during activation of immune cells and other cells under the influence of proinflammatory cytokines and chemokines. The B2M-free monomers give rise to homodimers [Face-3] and heterodimers [Face-4) on the cell surface. Although we have speculated a variety of dimers based on Figure-4, validation with further investigation is warranted. New reports are emerging on the functional diveristy of the dimers in association with other cell surface receptors such as KIR and the leukocyte Ig-like family of receptors. These diversified functions in normal cells and malignant cells merit extensive investigation, which may shed light on organ metastasis. Similarly, we have postulated that the B2M-free monomer could be the phylogenetic progenitor of HLA-Class I and class II, as well as of the B2M-free dimers. Only focused future investigations may either confirm or reject these postulated hypothesis.
